

# A perturbation theory for multi-time correlation functions in open quantum systems

**Piotr Szańkowski**

Institute of Physics, Polish Academy of Sciences,
al. Lotników 32/46, PL 02-668 Warsaw, Poland

piotr.szankowski@ifpan.edu.pl

## Abstract

*Dynamical maps* are the principal subject of the open system theory. Formally, the dynamical map of a given open quantum system is a density matrix transformation that takes any initial state and sends it to the state at a later time. Physically, it encapsulates the system's evolution due to coupling with its environment. Hence, the theory provides a flexible and accurate framework for computing expectation values of open system observables. However, expectation values—or more generally, single-time correlation functions—capture only the simplest aspects of a quantum system's dynamics. A complete characterization of the dynamics requires access to multi-time correlation functions as well: phenomena like detailed balance, linear and non-linear response, non-equilibrium transport in general, or even sequential measurements of system observables are all described in terms of multi-time correlations. For closed systems, such correlations are well-defined, even though knowledge of the system's state alone is insufficient to determine them fully. In contrast, the standard dynamical map formalism for open systems does not account for multi-time correlations, as it is fundamentally limited to describing state evolution. Here, we extend the scope of open quantum system theory by developing a systematic perturbation theory for computing multi-time correlation functions.



# 1  Introduction

Consider a two-level system, $q$, coupled to a thermal bath, $b$, at inverse temperature $\beta$. Initially, the qubit and the bath are uncorrelated, and their subsequent dynamics are governed by a simple Hamiltonian:

$$\hat{H}^{qb} = \hat{1}^q \otimes \hat{H}^b + \lambda \hat{\sigma}_x^q \otimes \hat{V}^b, \qquad \hat{\rho}_{t=0}^{qb} = \hat{\rho}_0^q \otimes \hat{\rho}_0^b = |1_z\rangle\langle 1_z| \otimes \hat{p}_\beta^b. \tag{1}$$

Here, $\hat{p}_\beta^b$ indicates the equilibrium state, which, in the case of the bath $b$, is given by

$$\hat{p}_\beta^b = \frac{\mathrm{e}^{-\beta \hat{H}^b}}{\mathrm{tr}_b\, \mathrm{e}^{-\beta \hat{H}^b}}, \tag{2}$$

the qubit's initial state was set to one of the eigenstates of the $z$-component of the spin operator, $\hat{\sigma}_z^q|\pm 1_z\rangle = \pm|\pm 1_z\rangle$, and for simplicity, we assume $\langle V^b \rangle = \mathrm{tr}(\hat{V}^b \hat{p}_\beta^b) = 0$.

Physical intuition suggests that interactions with the bath should eventually drive the qubit toward thermal equilibrium. Describing this process is a paradigmatic problem in open quantum system theory. Using its methods, one should be able to derive the dynamical map $\Lambda_t^q$ for the qubit state $\hat{\rho}_t^q = \mathrm{tr}_b(\hat{\rho}_t^{qb}) = \Lambda_t^q \hat{\rho}_0^q$, which models—approximately yet accurately—the dynamics of thermalization, $\Lambda_t^q \hat{\rho}_0^q \to \hat{p}_\beta^q$.

Arguably, the most powerful tool for addressing such problems in open quantum system theory is the renowned *master equation* method. It is based on the theorem proven by Gorini, Kossakowski, Lindblad, and Sudarshan (GKLS) [1,2] which provides an elegant *parametrization* of a class of dynamical maps. These maps are typically defined as a solution to the master equation—the namesake of the method—which takes the form

$$\frac{\mathrm{d}}{\mathrm{d}t}\Lambda_t = \mathcal{L}\Lambda_t \quad \Longrightarrow \quad \Lambda_t = \mathrm{e}^{t\mathcal{L}}. \tag{3}$$

The theorem asserts that the resulting map is physically viable provided its generator $\mathcal{L}$ has a well-defined and transparent structure—the GKLS form—composed of the so-called *jump operators*. Mathematically, jump operators are largely unrestricted; nevertheless, their contributions to the overall action of the map admit a clear physical interpretation.

The master equation method is applicable to the problem of thermalizing qubit in the weak-coupling regime,

$$\lambda \tau^b \ll 1, \tag{4}$$

where $\tau^b$ is the bath correlation time. In this regime, the evolution is accurately described by a special class of GKLS dynamical maps known as *Davis map*:

$$\hat{\rho}_t^q = \mathrm{e}^{-it[\hat{H}^q + \lambda\langle V^b\rangle\hat{\sigma}_x, \bullet] + t\mathcal{L}^q}\hat{\rho}_0^q = \mathrm{e}^{t\mathcal{L}^q}\hat{\rho}_0^q, \tag{5}$$

where the generator $\mathcal{L}^q$ takes the GKLS form and is derived from the original qubit–bath Hamiltonian:

$$\mathcal{L}^q = -\mathrm{i}h^b(0)\big[(\hat{\sigma}_x^q)^\dagger\hat{\sigma}_x^q, \bullet\big] + w^b(0)\Big(\hat{\sigma}_x^q\bullet\hat{\sigma}_x^q - \frac{1}{2}\{(\hat{\sigma}_x^q)^\dagger\hat{\sigma}_x^q, \bullet\}\Big) = w^b(0)\big(\hat{\sigma}_x^q\bullet\hat{\sigma}_x^q - \bullet\big), \tag{6}$$

with the *transition rate* and the *Lamb shift* given by

$$
\begin{aligned}
w^b(\omega) &= 2\lambda^2\,\mathrm{Re}\int_0^\infty \mathrm{tr}\Big[\big(\mathrm{e}^{\mathrm{i}\hat{H}^b(t+u)}\hat{V}^b\mathrm{e}^{-\mathrm{i}\hat{H}^b(t+u)}\big)\big(\mathrm{e}^{\mathrm{i}\hat{H}^bt}\hat{V}^b\mathrm{e}^{-\mathrm{i}\hat{H}^bt}\big)\hat{p}_\beta^b\Big]\mathrm{e}^{-\mathrm{i}\omega u}\mathrm{d}u \\
&= 2\lambda^2\,\mathrm{Re}\int_0^\infty \mathrm{tr}\Big[\mathrm{e}^{\mathrm{i}\hat{H}^bu}\hat{V}^b\mathrm{e}^{-\mathrm{i}\hat{H}^bu}\hat{V}^b\hat{p}_\beta^b\Big]\mathrm{e}^{-\mathrm{i}\omega u}\mathrm{d}u,
\end{aligned} \tag{7}
$$

$$h^b(\omega) = -2\lambda^2\,\mathrm{Im}\int_0^\infty \mathrm{tr}\Big[\mathrm{e}^{\mathrm{i}\hat{H}^bu}\hat{V}^b\mathrm{e}^{-\mathrm{i}\hat{H}^bu}\hat{V}^b\hat{p}_\beta^b\Big]\mathrm{e}^{-\mathrm{i}\omega u}\mathrm{d}u. \tag{8}$$

For such a simple coupling, the generator of the dynamical map contains only a single jump operator, $\hat{\sigma}_x^q$.

The first rigorous derivation of a GKLS-type map from the unitary dynamics of a system and its environment is attributed to E. B. Davies [3–7] (for a concise summary, see [8]). He proved that, under the assumption of a finite bath correlation time, the dynamical map in the *weak-coupling limit*—where the system–environment coupling strength $\lambda \to 0$ while the evolution duration $t \to \infty$ at a rate $\lambda^{-2}$—converges asymptotically to the Davies map. In this limit, the generator is uniquely determined by the system–environment Hamiltonian and the initial state of the environment, as exemplified in Eq. (6).

The intuitive expectation is that Davies' result should extend beyond the weak-coupling limit; it seems reasonable to presume that, under the right circumstances, the asymptotic Davies map could serve as an approximation to the exact dynamical map even for finite values of $\lambda$ and $t$. However, the weak-coupling limit itself does not provide rigorous justification for this expectation. Nevertheless, it strongly suggests that the weakness of the coupling is the key factor determining the accuracy of such an approximation.

The potential benefits of a reliable weak-but-finite coupling approximation justify relaxing strict mathematical rigor, provided that it can be compensated for with robust physical reasoning. At present, the standard approach—widely accepted in textbooks—is the *Born-Markov approximation* [9–12], which is built around the ansatz for the total density matrix:

$$\hat{\rho}_t^{qb} \approx \hat{\rho}_t^q \otimes \hat{p}_\beta^b. \tag{9}$$

A common justification for this approximation relies on the intuition that, for a sufficiently weak coupling, the perturbations induced in the state of a "large" bath should be negligible compared to their effect on the "small" open system. Alternatively, one can argue that any small deviation from equilibrium in the bath, caused by interactions with the open system, should naturally relax on a very fast timescale.

However, numerical studies [10] clearly demonstrate that, even when the Born–Markov approximation to the dynamical map is accurate, the Born–Markov ansatz remains a very poor approximation to the actual state of the total system. This raises doubts about how the ansatz should be interpreted. Some authors argue that it should never be used outside the specific context of deriving the Davies master equation, yet it should not be discarded outright, as there

are no better alternative routes to obtaining this approximation from first principles. Others treat the ansatz as a genuine approximation that can be used to draw conclusions about the total state of the system and its environment [13].

For now, we shall consider the Born–Markov method, with which we obtained (5), as a sound approximation. In Section 5, we will demonstrate an alternative derivation of the Davies master equation that does not rely on—or even comment on—the form of the total system–bath state.

Then, it is straightforward to show that the action of the Davies map will drive the qubit's initial state to one that coincides with the thermal equilibrium:

$$
e^{t\mathcal{L}^q}|1_z\rangle\langle 1_z| \xrightarrow{t\to\infty} \frac{1}{2}\hat{1}^q = \frac{e^{-\beta(\hat{H}^q+\lambda\hat{\sigma}^q_x\langle V^b\rangle)}}{\operatorname{tr} e^{-\beta(\hat{H}^q+\lambda\hat{\sigma}^q_x\langle V^b\rangle)}} = \hat{p}^q_\beta\,. \tag{10}
$$

In this case, the equilibrium state is proportional to the identity, as the energy levels of $q$ are degenerate ($\hat{H}^q = 0$).

A thermal bath is typically viewed as a complex system consisting of a large number of interacting subsystems, each at thermal equilibrium with a common inverse temperature $\beta$. The intuitive expectation is that once an external system, such as $q$, thermalizes, it effectively becomes another constituent subsystem of the bath.

If this physical picture holds—and there is little reason to doubt its general validity—then the assimilated system $q$ should have the capacity to act as a "vector" for the further "spread" of thermalization. In other words, since $q$ is now part of the bath, any other system that interacts with the bath through direct coupling to $q$ should equilibrate just as if it were interacting with one of the bath's original constituent subsystems.

Let $s$ be such a system which is brought into contact with the bath by coupling to one of the observables of $q$, say

$$
\hat{H}^{sqb} = \hat{H}^s \otimes \hat{1}^{qb} + \hat{1}^s \otimes \hat{H}^{qb} + \mu\hat{V}^s \otimes \hat{\sigma}^q_z \otimes \hat{1}^b\,, \qquad \hat{\rho}^{sqb}_{t=0} = \hat{\rho}^s_0 \otimes \hat{p}^{qb}_\beta\,, \tag{11}
$$

where $\hat{p}^{qb}_\beta \propto \exp(-\beta\hat{H}^{qb})$ is the thermal equilibrium state of qubit plus bath system. Assuming weak coupling ($\mu\tau^b \ll 1$), the state of $s$ evolves according to the corresponding Davies dynamical map,

$$
\hat{\rho}^s_t = e^{-it[\hat{H}^s+\mu\langle\sigma^q_z\rangle\hat{V}^s,\bullet]}e^{-it\sum_\omega h^q(\omega)[\hat{V}^s(-\omega)\hat{V}^s(\omega),\bullet]+t\sum_\omega w^q(\omega)\left(\hat{V}^s(\omega)\bullet\hat{V}^s(-\omega)-\frac{1}{2}\{\hat{V}^s(-\omega)\hat{V}^s(\omega),\bullet\}\right)}\hat{\rho}^s_0\,, \tag{12}
$$

where the jump operators are obtained from the frequency decomposition of the $s$-side coupling,

$$
\hat{V}^s(\omega) = \sum_{k,k'}\delta\left(H^s(k)-H^s(k')-\omega\right)|k\rangle\langle k|\hat{V}^s|k'\rangle\langle k'| = \left(\hat{V}^s(-\omega)\right)^\dagger\,, \tag{13}
$$

with respect to the renormalized energy levels of $s$,

$$
H^s, |k\rangle: \quad \left(\hat{H}^s + \mu\langle\sigma^q_z\rangle\hat{V}^s - H^s(k)\hat{1}^s\right)|k\rangle = 0\,. \tag{14}
$$

For $s$ to thermalize under these conditions, it is sufficient that the ratio between the transition rates $w^q(\omega)$ and $w^q(-\omega)$ in Eq. (12) obey the *detailed balance* [12],

$$
\frac{w^q(-\omega)}{w^q(\omega)} = e^{\beta_q\omega}\,. \tag{15}
$$

The positive coefficient $\beta_q$ determines the effective temperature of the equilibrium state asymptotically reached by $s$. Hence, if we believe the picture of spreading thermalization to be true, we expect $\beta_q$ to match the bath temperature $\beta$.

The transition rate is defined analogously to Eq. (7),

$$w^q(\omega) = 2\mu^2 \,\mathrm{Re} \int_0^\infty \mathrm{tr}\Big[ \mathrm{e}^{\mathrm{i}\hat{H}^{qb}u}(\hat{\sigma}_z^q - \langle\sigma_z^q\rangle\hat{1}^q)\otimes\hat{1}^b\, \mathrm{e}^{-\mathrm{i}\hat{H}^{qb}u}(\hat{\sigma}_z^q - \langle\sigma_z^q\rangle\hat{1}^q)\otimes\hat{1}^b\, \hat{p}_\beta^{qb}\Big]\mathrm{e}^{-\mathrm{i}\omega u}\mathrm{d}u\,. \quad (16)$$

Hence, to compute it, we require the *multi-time correlation* function of an observable belonging to the system *q open* to its environment *b*,

$$\begin{aligned}
\langle\sigma_z^q(t_2^+)\sigma_z^q(t_1^+)\rangle_{p_\beta^{qb}} &= \mathrm{tr}\Big[\big(\mathrm{e}^{\mathrm{i}\hat{H}^{qb}t_2}\hat{\sigma}_z^q\otimes\hat{1}^b\mathrm{e}^{-\mathrm{i}\hat{H}^{qb}t_2}\big)\big(\mathrm{e}^{\mathrm{i}\hat{H}^{qb}t_1}\hat{\sigma}_z^q\otimes\hat{1}^b\mathrm{e}^{-\mathrm{i}\hat{H}^{qb}t_1}\big)\hat{p}_\beta^{qb}\Big]\\
&= \mathrm{tr}\Big[\mathrm{e}^{\mathrm{i}\hat{H}^{qb}(t_2-t_1)}(\hat{\sigma}_z^q\otimes\hat{1}^b)\mathrm{e}^{-\mathrm{i}\hat{H}^{qb}(t_2-t_1)}(\hat{\sigma}_z^q\otimes\hat{1}^b)\hat{p}_\beta^{qb}\Big].
\end{aligned} \quad (17)$$

(The angle bracket notation convention will be explained in section 2.)

One would expect the theory of open systems to provide the means for deriving such correlation functions. However, as indicated previously, the theory's focus is on dynamical maps that describe the time evolution of the initial state of an open system. Consequently, the traditional implementation of the dynamical map method can be used only to obtain single-time correlation functions, i.e., expectation values of observables. Even when the correlations between system-only observables are of interest, the dynamical map is of little use in Eq. (17); simply put, multi-time correlations are beyond the scope of the standard formulation of the theory.

A common workaround for this problem is the *quantum regression formula* (QRF) [14–16]. In this approach, it is assumed that a correlation function for an *open* system can be approximated by modifying the form of an analogous correlation function for a *closed* system, by replacing instances of unitary propagators with dynamical maps. Symbolically, the QRF can be expressed as:

$$\mathrm{tr}\Big[\mathrm{e}^{\mathrm{i}\hat{H}t}\hat{V}\mathrm{e}^{-\mathrm{i}\hat{H}t}\hat{V}\hat{p}_\beta\Big] = \mathrm{tr}\Big[\hat{V}\big(\mathrm{e}^{-\mathrm{i}\hat{H}t}\bullet\mathrm{e}^{\mathrm{i}\hat{H}t}\big)(\hat{V}\hat{p}_\beta)\Big] \quad\overset{\mathrm{QRF}}{\longleftrightarrow}\quad \mathrm{tr}\Big[\hat{V}\mathrm{e}^{t\mathcal{L}}(\hat{V}\hat{p}_\beta)\Big]\,. \quad (18)$$

The motivation for this approximation is twofold: (i) it is the simplest generalization of the closed system formula, in which the unitary state evolution is replaced with the closest analogous transformation available for open systems; (ii) in the weak coupling limit, this form emerges asymptotically, with the dynamical maps identical to the Davies maps obtained in the same limit for single-time correlators [17].

Applying the QRF to our problem, we can compute the correlation function and obtain the following result:

$$\textbf{QRF:}\quad \langle\sigma_z^q(u^+)\sigma_z^q(0^+)\rangle_{p_\beta^{qb}} \approx \mathrm{tr}\Big[\hat{\sigma}_z^q\mathrm{e}^{u\mathcal{L}^q}(\hat{\sigma}_z^q\hat{p}_\beta^q)\Big] = \mathrm{e}^{-2w^b(0)u}\,, \quad (19)$$

which then leads to transition rates:

$$w^q(\omega) = 2\mu^2 \,\mathrm{Re}\int_0^\infty \mathrm{e}^{-2w^b(0)u}\mathrm{e}^{-\mathrm{i}\omega u}\mathrm{d}u = \frac{8\mu^2 w^b(0)}{(2w^b(0))^2+\omega^2} = w^q(-\omega)\,. \quad (20)$$

The resulting ratio of transition rates is unity, which does not technically violate the detailed balance (15). However, it implies an effective temperature of $\beta_q = 0$, which is inconsistent with the actual bath temperature $\beta$. Therefore, if we were to trust the QRF method, we would have to conclude that thermalization does not propagate, a result that contradicts our physical intuition.

Since the method itself is postulated rather than derived from a robust, physically sound principle, its veracity can only be judged by the results it produces. Fortunately, in this particular case, we are fairly confident that the obtained result is categorically incorrect (thermalization is believed to be a robust process with few exceptions), so there is no real risk of mistaking

this artifact of a flawed method for a genuine physical effect. However, the distinction might not always be so clear-cut—without proper theoretical underpinnings, it is impossible to estimate how much trust we should place in predictions made using the QRF method.

This example exposes an underappreciated blind spot in the theory of open quantum systems [18]: its reliance on the QRF for multi-time correlation functions is a significant weakness that severely limits the scope of the theory. To address this issue, we derive in this paper a comprehensive perturbation theory that supersedes the unreliable quantum regression formula method.

In particular, QRF is recovered in the zeroth-order of the perturbation theory, and, as demonstrated, the zeroth-order approximation fails to capture the propagation of thermalization accurately. However, in the case of our example, even the first-order correction suffices to recover the expected detailed balance (at least in the high-temperature regime, $\beta \ll \tau^b$),

$$
\begin{array}{r}
\textbf{Perturbation} \\
\textbf{theory} \\
\textbf{(1}^{\text{st}}\textbf{-order)}
\end{array}
\qquad
\frac{w^q(-\omega)}{w^q(\omega)} \approx \frac{1 + \beta\omega/2}{1 - \beta\omega/2} = 1 + \beta\omega + \frac{1}{2!}\beta^2\omega^2 + O(\beta^3\omega^3) \approx e^{\beta\omega} \,. \tag{21}
$$

The detailed derivation of the perturbation theory for multi-time correlations functions—as well as its specific implementation in the example discussed here—is presented in what follows. In Section 2, we provide the formal definition of the multi-time correlations (MTCs) and we introduce the bi-trajectory formalism for quantum mechanics, which will prove especially useful in describing MTCs in open systems. Section 3 describes the temporal correlations of bi-trajectories; we introduce moments and cumulants of bi-trajectory-dependent dynamical variables and use them to define the hierarchy of correlation strength—the key concept in the construction of our perturbation theory. In Section 4, we derive the quantum regression formula (QRF) approximation as the zeroth-order perturbation. Section 5 follows with a discussion of the analogue of the Born–Markov approximation as it applies to the MTC perturbation theory. Section 6 presents the explicit form of the first-order correction to the zeroth-order QRF. In Sec. 7, we return to the problem introduced above and apply the first-order perturbation theory to restore detailed balance. Conclusions and final remarks are collected in Sec. 8

## 2 Multi-time correlation functions and bi-trajectories

### 2.1 Multi-time correlation (MTC)

The general multi-time correlation (MTC) function for the ordered sequence of $n$ times and observables

$$
\mathbf{t}_n = (t_n, \ldots, t_1)\,, \qquad \mathbf{F}_n = (F_n, \ldots, F_1)\,, \tag{22}
$$

with $t_n \geqslant \cdots \geqslant t_1 \geqslant t_0 = 0$ and each $F_j$ represented by the corresponding operator $\hat{F}_j$ in the system's Hilbert space $\mathcal{H}$, under the initial $t = 0$ condition represented by a density matrix $\hat{\rho}$, is defined as:

$$
\begin{array}{r}
\textbf{Multi-time} \\
\textbf{correlation} \\
\textbf{(MTC)}
\end{array}
\qquad
\left\langle \prod_{j \in I^+} F_j(t_j^+) \prod_{k \in I^-} F_k(t_k^-) \right\rangle_\rho := \mathrm{tr}\left[ \mathrm{T}\left\{ \prod_{j \in I^+} \hat{F}_j(t_j) \right\} \hat{\rho} \, \mathrm{T}\left\{ \prod_{k \in I^-} \hat{F}_k(t_k) \right\}^\dagger \right]. \tag{23}
$$

Here, the indices $I_n = \{1, \ldots, n\}$ enumerating the times are distributed among sets $I^+$ and $I^-$ in such a way that $I^+ \cup I^- = I_n$. Time-dependent operators $\hat{F}_j(t)$ indicate the Heisenberg picture with respect to the unitary evolution operator describing the system's dynamical law,

$$
\hat{F}_j(t) = e^{i\hat{H}t}\, \hat{F}_j\, e^{-i\hat{H}t}\,, \tag{24}
$$

and $\mathrm{T}\{\cdots\}$ is the time-ordering operation, e.g., if $t_2 > t_1$, then for any $j, k$ one has

$$\mathrm{T}\left\{\hat{F}_j(t_2)\hat{F}_k(t_1)\right\} = \mathrm{T}\left\{\hat{F}_k(t_1)\hat{F}_j(t_2)\right\} = \hat{F}_j(t_2)\hat{F}_k(t_1). \tag{25}$$

Although the definition of MTC could, in principle, accommodate arbitrary operators $\hat{F}_j$, we shall consider only the case when all of them are hermitian, $\hat{F}_j^{\dagger} = \hat{F}_j$, so that each operator has valid spectral decomposition,

$$\hat{F}_j(t) = \sum_{f \in \Omega(F_j)} f\, \mathrm{e}^{\mathrm{i}\hat{H}t}\hat{P}^{F_j}(f)\mathrm{e}^{-\mathrm{i}\hat{H}t} \equiv \sum_{f \in \Omega(F_j)} f\, \hat{P}_t^{F_j}(f), \tag{26}$$

such that

$$\sum_{f \in \Omega(F_j)} \hat{P}_t^{F_j}(f) = \hat{1}, \qquad \hat{P}_t^{F_j}(f)\hat{P}_t^{F_j}(f') = \delta_{f,f'}\hat{P}_t^{F_j}(f), \tag{27}$$

where $\Omega(F_j)$ contains all unique eigenvalues, $\hat{F}_j\hat{P}^{F_j}(f) = f\hat{P}^{F_j}(f)$.

The restriction to proper hermitian observables does not curtail the generality of our considerations because any non-hermitian operator $\hat{A}$ can always be decomposed into a combination of hermitian operators,

$$\hat{A} = \hat{F}_{\mathrm{re}} + \mathrm{i}\hat{F}_{\mathrm{im}}: \quad \hat{F}_{\mathrm{re}} = \frac{\hat{A} + \hat{A}^{\dagger}}{2} = \hat{F}_{\mathrm{re}}^{\dagger}, \qquad \hat{F}_{\mathrm{im}} = \frac{\hat{A} - \hat{A}^{\dagger}}{2\mathrm{i}} = \hat{F}_{\mathrm{im}}^{\dagger}. \tag{28}$$

Therefore, MTC for non-hermitian operators can always be decomposed into a linear combination of MTCs for hermitian operators only.

## 2.2 Bi-trajectory formalism

Multi-time correlation functions are ostensibly *dynamical* quantities. As such, the formal description of MTCs in terms of the standard state-focused formulation of quantum mechanics—which is tailored for describing 'quantum statics' (i.e., single-time MTCs, or simply, expectation values of observables)—is a sub-optimal choice. Recently renewed interest in contexts extending beyond the standard quantum statics has stimulated development of powerful tools such as *process tensors* (or *quantum combs*) [19–21], that can efficiently handle various multi-time objects encountered in quantum mechanics. Here, our method of choice is the so-called *bi-trajectory formalism*; it was developed previously in [22–27] and it is a natural fit for the specific context of open system MTCs. The bi-trajectory theory is a reformulation of quantum mechanics that shares some of its DNA with other trajectory-focused formalisms [28–30]. Moreover, a limited preliminary investigation [25] suggests a deeply rooted connection between bi-trajectory formalism and process tensors, albeit the full understanding of this potentiality requires further study.

In bi-trajectory formalism, the dynamics of a collection of observables $\mathbf{F}_n$, given the initial condition $\hat{\rho}$, is described by a multi-time *bi-probability* distribution,

$$\begin{array}{c}\textbf{Bi-probability}\\ \textbf{distribution}\end{array} \quad Q_{\mathbf{t}_n|0}^{\mathbf{F}_n|\rho}(\mathbf{f}_n^+, \mathbf{f}_n^-) := \mathrm{tr}\left[\mathrm{T}\left\{\prod_{j=n}^{1}\hat{P}_{t_j}^{F_j}(f_j^+)\right\}\hat{\rho}\,\mathrm{T}\left\{\prod_{k=1}^{n}\hat{P}_{t_k}^{F_k}(f_k^-)\right\}^{\dagger}\right]. \tag{29}$$

Then, MTC is equivalent to a *moment* of bi-probability distribution,

$$\left\langle \prod_{j \in I^+} F_j(t_j^+) \prod_{k \in I^-} F_k(t_k^-) \right\rangle_{\rho} = \sum_{\mathbf{f}_n^{\pm} \in \Omega(\mathbf{F}_n)} \left(\prod_{j \in I^+} f_j^+ \prod_{k \in I^-} f_k^-\right) Q_{\mathbf{t}_n|0}^{\mathbf{F}_n|\rho}(\mathbf{f}_n^+, \mathbf{f}_n^-), \tag{30}$$

where we used the short-hand notation $\Omega(\mathbf{F}_n) = \Omega(F_n) \times \cdots \times \Omega(F_1)$ and $\mathbf{f}_n^{\pm} = (f_n^{\pm}, \ldots, f_1^{\pm})$ to represent $n$-element sequences of observable eigenvalues.

However, a multi-time bi-probability such as $Q^{\mathbf{F}_n|\rho}_{\mathbf{t}_n|0}$ is not the fundamental object of the formalism. Rather, it is a discrete-time restriction of a more general complex-valued *master distribution*,

$$\begin{matrix} \text{Master Bi-trajectory} \\ \text{Distribution} \end{matrix} \quad Q[\eta^+, \eta^-][D\eta^+][D\eta^-], \tag{31}$$

on the space of pairs of whole *trajectories*,

$$\textbf{Bi-trajectory} \quad \mathbb{R}_+ \times \mathbb{S}^{d^2-1} \ni (t, \vec{\varphi}) \mapsto (\eta^+(t, \vec{\varphi}), \eta^-(t, \vec{\varphi})) \in \{1, \dots, d\}^2. \tag{32}$$

Here, $d = \dim \mathcal{H}$ is the dimension of the system Hilbert space, $t$ is time, and coordinates $\vec{\varphi}$ on the $(d^2-1)$-dimensional hyper-sphere $\mathbb{S}^{d^2-1}$ parameterize unitary basis transformations,

$$\vec{\varphi} = \begin{bmatrix} \varphi^1 \\ \vdots \\ \varphi^{d^2-1} \end{bmatrix} : \quad \mathrm{e}^{-\mathrm{i}\vec{\varphi}\cdot\vec{\hat{T}}} = \exp\left( -\mathrm{i} \sum_{\ell=1}^{d^2-1} \varphi^\ell \hat{T}_\ell \right), \tag{33}$$

with $\{\hat{T}_\ell \mid \ell = 1, \dots, d^2-1\}$ the complete set of Hermitian generators of these transformations.

For closed quantum systems, the master distribution $Q[\eta^+, \eta^-]$ is generated by the system's dynamical law $\exp(-\mathrm{i}\hat{H}t)$ [27], and it describes all possible histories of the system's observables. Specifically, the multi-time bi-probabilities—and thus, MTCs—are derived from $Q[\eta^+, \eta^-]$ as follows:

$$Q^{\mathbf{F}_n|\rho}_{\mathbf{t}_n|t_0}(\mathbf{f}^+_n, \mathbf{f}^-_n) = \iint \Delta^{\mathbf{F}_n|\rho}_{\mathbf{t}_n|t_0}(\mathbf{f}^+_n, \mathbf{f}^-_n \mid \eta^+, \eta^-) Q[\eta^+, \eta^-][D\eta^+][D\eta^-], \tag{34}$$

where the discrete-time filter function

$$\Delta^{\mathbf{F}_n|\rho}_{\mathbf{t}_n|t_0}(\mathbf{f}^+_n, \mathbf{f}^-_n \mid \eta^+, \eta^-) = \Big( \sum_{\boldsymbol{\eta}^+_n} \prod_{j=1}^n \delta_{\eta^+_j, \eta^+(t_j, \vec{\varphi}_j)} \delta_{f^+_j, f_j(\eta^+_j)} \Big) \Big( \sum_{\boldsymbol{\eta}^-_n} \prod_{j=1}^n \delta_{\eta^-_j, \eta^-(t_j, \vec{\varphi}_j)} \delta_{f^-_j, f_j(\eta^-_j)} \Big)$$
$$\times \Big( \sum_{\eta_0} p(\eta_0) \delta_{\eta_0, \eta^+(t_0, \vec{\varphi}_0)} \delta_{\eta_0, \eta^-(t_0, \vec{\varphi}_0)} \Big), \tag{35}$$

imposes constraints on the components of bi-trajectory $\eta^\pm(t, \vec{\varphi})$ at the selected moments in time $\mathbf{t}_n, t_0$ and coordinates $\vec{\varphi}_n, \vec{\varphi}_0$. These coordinates correspond to the choices of observables $\mathbf{F}_n$ and the initial condition, and their concrete value is determined by the solution of the corresponding eigenproblem:

$$f_j, \vec{\varphi}_j : \ \big(\hat{F}_j - f_j(\eta)\hat{\mathbb{1}}\big)\mathrm{e}^{-\mathrm{i}\vec{\varphi}_j\cdot\vec{\hat{T}}}|\eta\rangle = 0, \qquad p, \vec{\varphi}_0 : \ \big(\hat{\rho} - p(\eta)\hat{\mathbb{1}}\big)\mathrm{e}^{-\mathrm{i}\vec{\varphi}_0\cdot\vec{\hat{T}}}|\eta\rangle = 0, \tag{36}$$

where $\{|\eta\rangle \mid \eta = 1, \dots, d\}$ is an arbitrarily chosen reference orthonormal basis. Equivalently, the correspondence can be expressed through the spectral decomposition of operators representing observables,

$$\hat{F}_j = \sum_{\eta=1}^d f_j(\eta) \mathrm{e}^{-\mathrm{i}\vec{\varphi}_j\cdot\vec{\hat{T}}}|\eta\rangle\langle\eta|\mathrm{e}^{\mathrm{i}\vec{\varphi}_j\cdot\vec{\hat{T}}}, \qquad \hat{\rho} = \sum_{\eta=1}^d p(\eta) \mathrm{e}^{-\mathrm{i}\vec{\varphi}_0\cdot\vec{\hat{T}}}|\eta\rangle\langle\eta|\mathrm{e}^{\mathrm{i}\vec{\varphi}_0\cdot\vec{\hat{T}}}. \tag{37}$$

Admittedly, switching to bi-trajectory picture might seem an unnecessarily complex solution to a relatively simple problem of computing MTCs in *closed* systems. However, once the system is open to its environment, and the standard state-focused formalism ceases to be practical, the formal description in terms of bi-trajectory becomes indispensable. The crucial feature of this formulation, which we will rely on in our derivations, is that the bi-trajectory $(\eta^+(t, \vec{\varphi}), \eta^-(t, \vec{\varphi}))$ provides a systematic way to track temporal correlations between observables.

## 2.3 Multi-time correlations in bi-trajectory formalism: Examples

Examples of physically relevant MTCs include the *auto-correlation function* $\mathrm{C}^F$ and the *susceptibility* $\mathrm{K}^F$ of an observable $F$, given by

$$
\begin{aligned}
\mathrm{C}^F_{t_2,t_1} &= \frac{1}{2}\mathrm{tr}\left(\{\hat{F}(t_2),\hat{F}(t_1)\}\hat{\rho}\right) - \mathrm{tr}\left(\hat{F}(t_2)\hat{\rho}\right)\mathrm{tr}\left(\hat{F}(t_1)\hat{\rho}\right) \\
&= \mathrm{Re}\,\mathrm{tr}\left[\left(\hat{F}(t_2)-\mathrm{tr}\left[\hat{F}(t_2)\hat{\rho}\right]\right)\left(\hat{F}(t_1)-\mathrm{tr}\left[\hat{F}(t_1)\hat{\rho}\right]\right)\hat{\rho}\right] \\
&= \mathrm{Re}\langle F(t_2^+)F(t_1^+)\rangle_\rho - \langle F(t_2^+)\rangle_\rho\langle F(t_1^+)\rangle_\rho, \quad\quad (38a)
\end{aligned}
$$

$$
\begin{aligned}
\mathrm{K}^F_{t_2,t_1} &= \Theta(t_2-t_1)\frac{1}{2\mathrm{i}}\mathrm{tr}\left([\hat{F}(t_2),\hat{F}(t_1)]\hat{\rho}\right) \\
&= \Theta(t_2-t_1)\mathrm{Im}\,\mathrm{tr}\left[\left(\hat{F}(t_2)-\mathrm{tr}\left[\hat{F}(t_2)\hat{\rho}\right]\right)\left(\hat{F}(t_1)-\mathrm{tr}\left[\hat{F}(t_1)\hat{\rho}\right]\right)\hat{\rho}\right] \\
&= \Theta(t_2-t_1)\mathrm{Im}\langle F(t_2^+)F(t_1^+)\rangle_\rho, \quad\quad (38b)
\end{aligned}
$$

where $\{\hat{A},\hat{B}\}=\hat{A}\hat{B}+\hat{B}\hat{A}$, $[\hat{A},\hat{B}]=\hat{A}\hat{B}-\hat{B}\hat{A}$ and $\Theta(t)=1$ if $t>0$ or 0 otherwise.

These two-time functions play a crucial role in many physical contexts; here, we highlight one significant example. When the initial state is thermal equilibrium, $\hat{\rho}=\hat{p}_\beta \propto \exp(-\beta\hat{H})$, the auto-correlation function and susceptibility appear on opposite sides of the well-known fluctuation–dissipation theorem [12, 31, 32]:

$$
\mathrm{Im}\int_{-\infty}^{\infty}\mathrm{K}^F_{t,0}e^{-\mathrm{i}\omega t}\mathrm{d}t = \left(\frac{1-e^{\beta\omega}}{1+e^{\beta\omega}}\right)\int_{-\infty}^{\infty}\mathrm{C}^F_{t,0}e^{-\mathrm{i}\omega t}\mathrm{d}t. \quad\quad (39)
$$

The very specific interplay between spectral decompositions of auto-correlation and susceptibility is what causes the detailed balance relation (15) between the transition rates in the thermalization process discussed in section 1 [12].

The expectation value of the observable $F$ appearing in the auto-correlation function $\mathrm{C}^F$ is a simple single-time MTC, which corresponds to the first moment of the associated bi-probability distribution:

$$
\langle F(t^+)\rangle_\rho = \sum_{f_1^\pm} f_1^+ Q^{F|\rho}_{t|0}(f_1^+;f_1^-) = \sum_{f_1^\pm} f_1^- Q^{F|\rho}_{t|0}(f_1^+;f_1^-) = \langle F(t^-)\rangle_\rho. \quad\quad (40)
$$

Accordingly, the two-time MTC, which determines both $\mathrm{C}^F$ and $\mathrm{K}^F$, is given by the second moment:

$$
\langle F(t_2^+)F(t_1^+)\rangle_\rho = \sum_{\mathbf{f}_2^\pm} f_2^+ f_1^+ Q^{F|\rho}_{\mathbf{t}_2|0}(\mathbf{f}_2^+,\mathbf{f}_2^-) = \sum_{\mathbf{f}_2^\pm} f_2^- f_1^- Q^{F|\rho}_{\mathbf{t}_2|0}(\mathbf{f}_2^+,\mathbf{f}_2^-)^* = \langle F(t_1^-)F(t_2^-)\rangle_\rho^*, \quad (41)
$$

and thus

$$
\mathrm{Re}\langle F(t_2^+)F(t_1^+)\rangle_\rho = \frac{1}{2}\sum_{\mathbf{f}_2^\pm}\left(f_2^+ f_1^+ + f_1^- f_2^-\right)Q^{F|\rho}_{\mathbf{t}_2|0}(\mathbf{f}_2^+,\mathbf{f}_2^-), \quad\quad (42)
$$

$$
\mathrm{Im}\langle F(t_2^+)F(t_1^+)\rangle_\rho = \frac{1}{2\mathrm{i}}\sum_{\mathbf{f}_2^\pm}\left(f_2^+ f_1^+ - f_1^- f_2^-\right)Q^{F|\rho}_{\mathbf{t}_2|0}(\mathbf{f}_2^+,\mathbf{f}_2^-). \quad\quad (43)
$$

## 3 Temporal correlations of bi-trajectory

A key concept for our derivation of the perturbation theory is the hierarchy of bi-trajectory temporal correlation strength: the order of perturbation will be defined with respect to this hierarchy.

Consider the $n$-time MTC for certain sequence of timings $\mathbf{t}_n$ and observables $\mathbf{F}_n$,

$$\Big\langle \prod_{i\in I^+} F_j(t_j^+) \prod_{k\in I^-} F_k(t_k^-) \Big\rangle_\rho = \sum_{\mathbf{f}_n^\pm} \Big( \prod_{j\in I^+} f_j^+ \prod_{k\in I^-} f_k^- \Big) Q_{\mathbf{t}_n|0}^{\mathbf{F}_n|\rho}(\mathbf{f}_n^+, \mathbf{f}_n^-). \tag{44}$$

Assuming $I^+ \cap I^- = \emptyset$, we introduce the shorthand notation that focuses only on the temporal arrangement of observables:

$$\Big\langle \prod_{i\in I^+} F_j(t_j^+) \prod_{k\in I^-} F_k(t_k^-) \Big\rangle_\rho \equiv \langle Z_n \cdots Z_1 \rangle. \tag{45}$$

Hence, $Z_i = F_i(t_i^\pm)$ depending on whether $i \in I^+$ or $i \in I^-$.

For a sufficiently complex system, such as a thermal bath, it is expected on physical grounds that temporal correlations between system observables—represented here by *dynamical variables $Z_j$*—have *finite range*. This characteristic time-scale on which dynamical variables become independent, is called the *correlation time $\tau$*. Here, we employ moments of bi-probabilities (i.e., the MTCs) to define $\tau$ for the system under consideration:

$$\textbf{Correlation time } \tau: \quad \langle \cdots Z_{r+1} Z_r \cdots \rangle \xrightarrow{t_{r+1} - t_r > \tau} \langle \cdots Z_{r+1} \rangle \langle Z_r \cdots \rangle. \tag{46}$$

In words: If it is finite, the correlation time $\tau$ is a time-scale on which the underlying bi-trajectory decorrelates. Therefore, when we take any $n$-time moment of bi-probability distribution, and we split its timings into two blocks, say $(t_n, \ldots, t_{r+1})$ and $(t_r, \ldots, t_1)$, it factorizes into $(n-r+1)$- and $r$-time moments when the delay between blocks is greater than $\tau$.

Moments such as (45) are just one way of characterizing a distribution. Although moments are straightforward to calculate, they are inefficient for quantifying correlations of dynamical variables. The reason is that moments are *statistically reducible*, meaning that a moment of order $n$ contains unique information about $n$-time correlations along with redundant information about lower-order correlations found in other moments. If correlations are the focus, then the standard practice is to employ quantities that are statistically irreducible—the *cumulants*. Conventionally, cumulants $\langle\langle Z_n \cdots Z_1 \rangle\rangle$ are defined using moments:

$$\textbf{Cumulants } \langle\langle \cdots \rangle\rangle: \quad \langle Z_n \cdots Z_1 \rangle \equiv \sum_{P \in \text{Part}\{n,\ldots,1\}} \prod_{\mathbf{p}_k \in P} \langle\langle Z_{p_k} \cdots Z_{p_1} \rangle\rangle. \tag{47}$$

Here, the sum runs over elements $P$ of the set of ordered partitions of index set $\{n, \ldots, 1\}$, $\text{Part}\{n, \ldots, 1\}$, e.g.,

$$\text{Part}\{3, 2, 1\} = \Big\{ \{(3,2,1)\}, \{(3,2),(1)\}, \{(3,1),(2)\}, \{(2,1),(3)\}, \{(3),(2),(1)\} \Big\}.$$

The subsequent product then runs over sequences $\mathbf{p}_k = (p_k, \ldots, p_1) \in P$, for example: if $P = \{(3,1),(2)\}$, then we have $P \ni \mathbf{p}_2 = (3,1)$ with components $p_2 = 3$ and $p_1 = 1$, and $P \ni \mathbf{p}_1 = (2)$ with a single component $p_1 = 2$.

One can iterate the definition (47) over $n$ to express a cumulant of any order in terms of moments; for example, the first three cumulants are given by:

$$\langle\langle Z_1 \rangle\rangle = \langle Z_1 \rangle,$$
$$\langle\langle Z_2 Z_1 \rangle\rangle = \langle Z_2 Z_1 \rangle - \langle Z_2 \rangle \langle Z_1 \rangle,$$
$$\langle\langle Z_3 Z_2 Z_1 \rangle\rangle = \langle Z_3 Z_2 Z_1 \rangle + 2\langle Z_3 \rangle \langle Z_2 \rangle \langle Z_1 \rangle - \langle Z_3 \rangle \langle Z_2 Z_1 \rangle - \langle Z_3 Z_2 \rangle \langle Z_1 \rangle - \langle Z_3 Z_1 \rangle \langle Z_2 \rangle.$$

The original definition of the correlation time in terms of factorizing moments can be recast in a more elegant and, ultimately, more useful form using cumulants,

$$\textbf{Correlation time } \tau: \quad \langle\langle \cdots Z_j \cdots Z_k \cdots \rangle\rangle \xrightarrow{t_j - t_k > \tau} 0. \tag{48}$$

This follows from irreducibility of cumulants: for a cumulant to be non-zero, every dynamical variable $Z_r$ must be correlated with all other $Z_j$'s, therefore, whenever any pair of variables decorrelates for whatever reason, the cumulant vanishes.[1]

We can now combine moments with cumulants to define the hierarchy of correlation strength between blocks of dynamical variables. We start by splitting the sequence of timings $\mathbf{t}_n$ into two blocks $(t_n, \ldots, t_{r+1})$ and $(t_r, \ldots, t_1)$; then we take a $n$-time moment and using the relation (47) we decompose it as follows:

**Hierarchy of Correlation Strength**

$$
\begin{aligned}
\langle Z_n \cdots Z_{r+1} Z_r \cdots Z_1 \rangle = {} & \langle Z_n \cdots Z_{r+1} \rangle \langle Z_r \cdots Z_1 \rangle \\
& + \sum_{j=r+1}^{n} \sum_{k=1}^{r} \langle Z_n \cdots \cancel{Z_j} \cdots Z_{r+1} \rangle \langle Z_r \cdots \cancel{Z_k} \cdots Z_1 \rangle \langle\langle Z_j Z_k \rangle\rangle \\
& + \sum_{j,j'=r+1}^{n} \sum_{k,k'=1}^{r} \langle \cdots \cancel{Z_j} \cdots \cancel{Z_{j'}} \cdots \rangle \langle \cdots \cancel{Z_k} \cdots \cancel{Z_{k'}} \cdots \rangle \\
& \quad \times \langle\langle Z_j Z_k \rangle\rangle \langle\langle Z_{j'} Z_{k'} \rangle\rangle \\
& \vdots \\
& + \sum_{i=r+1}^{n} \sum_{j=1}^{r} \langle Z_i \rangle \langle Z_j \rangle \langle\langle Z_n \cdots \cancel{Z_i} \cdots \cancel{Z_j} \cdots Z_1 \rangle\rangle \\
& + \sum_{i=1}^{n} \langle Z_i \rangle \langle\langle Z_n \cdots \cancel{Z_i} \cdots Z_1 \rangle\rangle \\
& + \langle\langle Z_n \cdots Z_1 \rangle\rangle \, .
\end{aligned}
\tag{49}
$$

As the blocks of timings get delayed with respect to each other, $t_{r+1} - t_r \to \tau$, the correlations between the blocks degrade. The expectation is that the higher the order of the cumulant, the faster its decay, because it involves larger number of variables from both blocks that decorrelate simultaneously. Therefore, as the delay approaches the correlation time, the lowest-order correction to completely independent blocks (and thus, factorized moments) consists of the two-time cumulant terms:

$$
\begin{aligned}
\langle Z_n \cdots Z_{r+1} Z_r \cdots Z_1 \rangle \xrightarrow{t_{r+1} - t_r \gtrsim \tau} {} & \langle Z_n \cdots Z_{r+1} \rangle \langle Z_r \cdots Z_1 \rangle \\
& + \sum_{j=r+1}^{n} \sum_{k=1}^{r} \langle Z_n \cdots \cancel{Z_j} \cdots Z_{r+1} \rangle \langle Z_r \cdots \cancel{Z_k} \cdots Z_1 \rangle \langle\langle Z_j Z_k \rangle\rangle \, .
\end{aligned}
\tag{50}
$$

In what follows we will not be considering correction of order higher than the second cumulant. We can then adopt a convenient specialized notation:

$$
\langle \cdots \overbrace{Z_j} \cdots \rangle \langle \cdots \overbrace{Z_k} \cdots \rangle \longleftrightarrow \langle \cdots \cancel{Z_j} \cdots \rangle \langle \cdots \cancel{Z_k} \cdots \rangle \langle\langle Z_j Z_k \rangle\rangle \, .
\tag{51}
$$

This Wick-type notation convention becomes necessary in the case when dynamical variables are non-commuting operators rather than numbers, and thus, the placement of variables in their respective blocks has to be preserved.

---

[1] We warn against misconstruing short correlation time with "short memory", or any other notion of Markovianity encountered in the literature. In general, the correlation's range is largely independent of Markovianity. Helpful rule of thumb is to think of Markovianity as a characterization of the underlying dynamical laws (e.g., are the governing equations of motion local in time?) and correlation functions (or cumulants) as describing regularities in the process realizations (e.g., given the process at $t = 0$, how long one has to wait for the process recurrence?). Classic example illustrating the difference is a deterministic processes describing planet orbiting its star: the process representing the planet's position is Markovian ('have no memory') since the generating dynamical equations are time-local, and, simultaneously, the correlation time is infinitely long because, due to orbital motion, correlation functions for planet's position exhibit oscillatory behavior.

# 4 Quantum regression formula as the zeroth-order perturbation

Let us consider a generic case of a quantum system $s$ interacting with its environment $e$,

$$\hat{H}^{se} = \hat{H}^s_0 \otimes \hat{1}^e + \hat{1}^s \otimes \hat{H}^e + \lambda \sum_\alpha \hat{V}^s_\alpha \otimes \hat{V}^e_\alpha$$

$$= \hat{H}^s \otimes \hat{1}^e + \hat{1}^s \otimes \hat{H}^e + \lambda \sum_\alpha \hat{V}^s_\alpha \otimes \hat{E}_\alpha \, , \tag{52}$$

$$\hat{\rho}^{se}_{t_0=0} = \hat{\rho}^s_0 \otimes \hat{\rho}^e_0 \, , \tag{53}$$

where $\lambda$ is the overall interaction strength and we have introduced the renormalized system Hamiltonian and environment couplings:

$$\hat{H}^s = \hat{H}^s_0 + \lambda \sum_\alpha \text{tr}(\hat{V}^e_\alpha \hat{\rho}^e_0) \hat{V}^s_\alpha \, , \qquad \hat{E}_\alpha = \hat{V}^e_\alpha - \text{tr}(\hat{V}^e_\alpha \hat{\rho}^e_0) \hat{1}^e \, . \tag{54}$$

For simplicity, we also assume stationary initial state of $e$, $[\hat{H}^e, \hat{\rho}^e_0] = 0$.

Formally, the dynamical properties of the total system are described by the master distribution $Q^{se}$ of $s$ and $e$ bi-trajectories that is generated by the dynamical law of the composite system $\exp(-it\hat{H}^{se})$. However, if only the system $s$ is of interests, only the distribution of $(\eta^+, \eta^-)$ is required, which can be obtained using the law of total measure [27]:

$$Q^s[\eta^+, \eta^-] = \iint Q^{s|e}[\eta^+, \eta^- \mid \varepsilon^+, \varepsilon^-] Q^e[\varepsilon^+, \varepsilon^-][D\varepsilon^+][D\varepsilon^-] \, . \tag{55}$$

Here, the conditional distribution $Q^{s|e}$ is determined by $\hat{H}^s$ and the form of the system-environment coupling and, crucially, $Q^e$ is the master distribution for bi-trajectory of *free* environment, i.e., $Q^e$ is generated by the dynamical law of $e$ alone, $\exp(-it\hat{H}^e)$.

In particular, our goal is to compute MTCs for sequence of $s$-only observables $\mathbf{F}^s_n$, such that each $F^s_j$ is represented by operator of form $\hat{F}^s_j \otimes \hat{1}^e$. These correlation functions are equal to moments of the multi-time bi-probability distributions which are, in turn, obtained from the reduced master distribution $Q^s$; the explicit form of those bi-probabilities is derived in appendix A (see also [24, 27]), and it reads:

$$\begin{array}{l} \textbf{Open System} \\ \textbf{Bi-probability} \end{array} \quad Q^{\mathbf{F}^s_n|\rho^{se}}_{\mathbf{t}_n|0}(\mathbf{f}^+_n, \mathbf{f}^-_n) = \text{tr}_s\left[\left\langle \prod_{j=n}^1 \mathcal{P}_j \, \text{T}\, e^{-i\lambda \int_{t_{j-1}}^{t_j} \mathcal{Z}_u du} \right\rangle^e \hat{\rho}^s_0\right], \tag{56}$$

where we have introduced:

1. The super-operator dynamical variable dependent on the environment bi-trajectory,

$$\begin{array}{l} \textbf{Dynamical} \\ \textbf{Variable} \end{array} \quad \mathcal{Z}_t = \sum_\alpha \sum_{\varepsilon^\pm_\alpha} \left(e_\alpha(\varepsilon^+_\alpha)\hat{V}^s_\alpha(t)\bullet - \bullet \hat{V}^s_\alpha(t)e_\alpha(\varepsilon^-_\alpha)\right)\delta_{\varepsilon^+_\alpha, \varepsilon^+(t,\vec{\varphi}_\alpha)}\delta_{\varepsilon^-_\alpha, \varepsilon^-(t,\vec{\varphi}_\alpha)}, \tag{57}$$

with the basis transformation coordinates $\vec{\varphi}_\alpha$ determined as the solution to the eigenproblem for the corresponding operator in $e$,

$$e_\alpha, \vec{\varphi}_\alpha : \, \left(\hat{E}_\alpha - e_\alpha(\varepsilon)\hat{1}\right)e^{-i\vec{\varphi}_\alpha \cdot \vec{\hat{T}}^e}|\varepsilon\rangle = 0 \, , \tag{58}$$

equivalently, $\vec{\varphi}_\alpha$ and $e_\alpha$ are such that

$$\hat{E}_\alpha = \sum_\varepsilon e_\alpha(\varepsilon)e^{-i\vec{\varphi}_\alpha \cdot \vec{\hat{T}}^e}|\varepsilon\rangle\langle\varepsilon|e^{i\vec{\varphi}_\alpha \cdot \vec{\hat{T}}^e} \, . \tag{59}$$

The interaction picture of operators in $s$ is defined with respect to the renormalized system Hamiltonian,

$$\hat{V}^s_\alpha(t) = e^{i\hat{H}^s t}\hat{V}^s_\alpha e^{-i\hat{H}^s t} = e^{it\left(\hat{H}^s_0 + \lambda \sum_{\alpha'} \text{tr}[\hat{V}^e_{\alpha'}\hat{\rho}^e_0]\hat{V}^s_{\alpha'}\right)}\hat{V}^s_\alpha e^{-it\left(\hat{H}^s_0 + \lambda \sum_{\alpha'} \text{tr}[\hat{V}^e_{\alpha'}\hat{\rho}^e_0]\hat{V}^s_{\alpha'}\right)} \, . \tag{60}$$

2. The 'bi-average' over the environmental bi-trajectory,

**Bi-average** $\quad \langle \cdots \rangle^e = \iint (\cdots) \Big( \sum_{\varepsilon_0} p^e(\varepsilon_0) \delta_{\varepsilon_0, \varepsilon^+(0, \vec{\varphi}_0)} \delta_{\varepsilon_0, \varepsilon^-(0, \vec{\varphi}_0)} \Big) Q^e[\varepsilon^+, \varepsilon^-][D\varepsilon^+][D\varepsilon^-],$

$$(61)$$

with $p^e$ and $\vec{\varphi}_0$ defined by $(\hat{\rho}^e - p^e(\varepsilon)) \exp(-i\vec{\varphi}_0 \cdot \vec{\hat{T}}^e)|\varepsilon\rangle = 0$, or

$$\hat{\rho}^e = \sum_{\varepsilon} p^e(\varepsilon) e^{-i\vec{\varphi}_0 \cdot \vec{\hat{T}}^e} |\varepsilon\rangle \langle\varepsilon| e^{i\vec{\varphi}_0 \cdot \vec{\hat{T}}^e}. \tag{62}$$

3. The shorthand notation for the interspersing 'interventions',

**Intervention** $\quad \begin{aligned} \mathcal{P}_j &:= \hat{P}_{t_j}^{F_j^s}(f_j^+) \bullet \hat{P}_{t_j}^{F_j^s}(f_j^-) \\ &= \big( e^{i\hat{H}^s t_j} \bullet e^{-i\hat{H}^s t_j} \big) \big( \hat{P}^{F_j^s}(f_j^+) \bullet \hat{P}^{F_j^s}(f_j^-) \big) \big( e^{-i\hat{H}^s t_j} \bullet e^{i\hat{H}^s t_j} \big). \end{aligned} \tag{63}$

Then, the quantum regression formula approximation is recovered by neglecting any temporal correlations of bi-trajectory $(\varepsilon^+, \varepsilon^-)$ between the time-segments separated by interventions: First, we expand the time-ordered exp—the *propagators*—into series,

**Propagator** $\quad \mathrm{T}\,e^{-i\lambda \int_{t'}^t \mathcal{Z}_u du} = \sum_{k=0}^{\infty} (-i\lambda)^k \int_{t'}^t du_k \cdots \int_{t'}^{u_2} du_1 \mathcal{Z}_{u_n} \cdots \mathcal{Z}_{u_1}, \tag{64}$

resulting in the bi-probability decomposed into a combination of moments of dynamical variables $\mathcal{Z}_t$

$$\begin{aligned} Q_{\mathbf{t}_n|0}^{\mathbf{F}_n^s|\rho^{se}}(\mathbf{f}_n^+, \mathbf{f}_n^-) = &\sum_{k_n=0}^{\infty} \cdots \sum_{k_1=0}^{\infty} \left( (-i\lambda)^{k_n} \int_{t_{n-1}}^{t_n} du_{k_n}^n \cdots \int_{t_{n-1}}^{u_1^n} du_1^n \right) \cdots \left( (-i\lambda)^{k_1} \int_{t_0}^{t_1} du_{k_1}^1 \cdots \int_{t_0}^{u_1^1} du_1^1 \right) \\ &\times \mathrm{tr}_s \Big[ \big\langle (\mathcal{P}_n \mathcal{Z}_{u_{k_n}^n} \cdots \mathcal{Z}_{u_1^n}) \cdots (\mathcal{P}_2 \mathcal{Z}_{u_{k_2}^2} \cdots \mathcal{Z}_{u_1^2})(\mathcal{P}_1 \mathcal{Z}_{u_{k_1}^1} \cdots \mathcal{Z}_{u_1^1}) \big\rangle^e \hat{\rho}_0^s \Big]. \end{aligned} \tag{65}$$

Afterwards, we apply to each of those moments the decomposition (49) into blocks consisting of timings $(u_{k_j}^j, \ldots, u_1^j)$ between pairs of interventions $\mathcal{P}_j$ and $\mathcal{P}_{j-1}$, while neglecting *all* cumulants; the result reads

$$\begin{aligned} Q_{\mathbf{t}_n|0}^{\mathbf{F}_n^s|\rho^{se}}(\mathbf{f}_n^+, \mathbf{f}_n^-) &\approx \mathrm{tr}_s \Big[ \Big( \prod_{j=n}^1 \mathcal{P}_j \sum_{k=0}^{\infty} (-i\lambda)^k \int_{t_{j-1}}^{t_j} du_k \cdots \int_{t_{j-1}}^{u_2} du_1 \langle \mathcal{Z}_{u_k} \cdots \mathcal{Z}_{u_1} \rangle^e \Big) \hat{\rho}_0^s \Big] \\ &= \mathrm{tr}_s \Big[ \Big( \prod_{j=n}^1 \mathcal{P}_j \big\langle \mathrm{T}\,e^{-i\lambda \int_{t_{j-1}}^{t_j} \mathcal{Z}_u du} \big\rangle^e \Big) \hat{\rho}_0^s \Big]. \end{aligned} \tag{66}$$

Since we have neglected every cumulant, this is the zeroth-order approximation with respect to the hierarchy of correlations strength. Moreover, this approximation is consistent with the weak-coupling regime, $\lambda \tau^e \ll 1$, where $\tau^e$ is the correlation time of environment. To show this we need to estimate the order of magnitude of first-order perturbation (with respect to the hierarchy of correlation strength) due to bi-trajectory correlations connecting dynamical variables *across* the interventions; such a perturbation originate from the terms of the form

$$\cdots \rangle^e \mathcal{P}_{j+1} \lambda^2 \int_{t_j}^{u_{r+1}^j} du_r^j \int_{t_{j-1}}^{u_{r'+1}^{j-1}} du_{r'}^{j-1} \langle \mathcal{Z}_{u_{k_j}^j} \cdots \overline{\mathcal{Z}_{u_r^j} \cdots \mathcal{Z}_{u_1^j}}\rangle^e \mathcal{P}_j \langle \mathcal{Z}_{u_{k_{j-1}}^{j-1}} \cdots \overline{\mathcal{Z}_{u_{r'}^{j-1}} \cdots \mathcal{Z}_{u_1^{j-1}}}\rangle^e \mathcal{P}_{j-1} \langle \cdots,$$

with the cumulant of the two super-operator dynamical variables that evaluates to:

$$
\cdots \overline{\mathcal{Z}_{u_2} \cdots \mathcal{Z}_{u_1}} \cdots = \sum_{\alpha_2, \alpha_1} \Big( \langle E_{\alpha_2}(u_2^+) E_{\alpha_1}(u_1^+) \rangle_{\rho^e} \cdots (\hat{V}^s_{\alpha_2}(u_2)\bullet) \cdots (\hat{V}^s_{\alpha_1}(u_1)\bullet) \cdots
$$
$$
+ \langle E_{\alpha_2}(u_2^-) E_{\alpha_1}(u_1^-) \rangle_{\rho^e} \cdots (\bullet \hat{V}^s_{\alpha_2}(u_2)) \cdots (\bullet \hat{V}^s_{\alpha_1}(u_1)) \cdots \tag{67}
$$
$$
+ \langle E_{\alpha_2}(u_2^+) E_{\alpha_1}(u_1^-) \rangle_{\rho^e} \cdots (\hat{V}^s_{\alpha_2}(u_2)\bullet) \cdots (\bullet \hat{V}^s_{\alpha_1}(u_1)) \cdots
$$
$$
+ \langle E_{\alpha_2}(u_2^-) E_{\alpha_1}(u_1^+) \rangle_{\rho^e} \cdots (\bullet \hat{V}^s_{\alpha_2}(u_2)) \cdots (\hat{V}^s_{\alpha_1}(u_1)\bullet) \cdots \Big).
$$

Assuming long delay between interventions, $t_j - t_{j-1} \gg \tau^e$, and given that cumulants have finite range $\tau^e$, the first-order perturbation roughly scales as

$$
\lambda^2 \int_{t_j}^{u_{r+1}^j} \mathrm{d}u_r^j \int_{t_{j-1}}^{u_{r'+1}^{j-1}} \mathrm{d}u_{r'}^{j-1} \cdots \overline{\mathcal{Z}_{u_r^j} \cdots \mathcal{P}_j \cdots \mathcal{Z}_{u_{r'}^{j-1}}} \cdots \sim (\lambda \tau^e)^2. \tag{68}
$$

Hence, the subsequent orders of the perturbation theory can be considered as 'small corrections' to the zeroth-order QRF, only if the system–environment coupling is weak. In other words, the weak-coupling regime $\lambda \tau^e \ll 1$ is a necessary condition for the QRF to be a valid approximation.

## 5 Born–Markov approximation to propagators

The zeroth-order of the proposed MTC perturbation theory neglects all temporal correlations of the environmental bi-trajectory that would reach across the interventions and connect propagators operating in between them. Higher-order terms in the perturbation series would then incorporate such connections, increasing in number and complexity as the series descends the hierarchy of correlation strength.

However, the temporal bi-trajectory correlations are also present within the intervals in between interventions, where they connect dynamical variables residing inside propagators themselves:

$$
\Big\langle \mathrm{T} e^{-i\lambda \int_{t_{j-1}}^{t_j} \mathcal{Z}_u \mathrm{d}u} \Big\rangle^e = \sum_{n=0}^{\infty} (-i\lambda)^n \int_{t_{j-1}}^{t_j} \mathrm{d}u_n \cdots \int_{t_{j-1}}^{u_2} \mathrm{d}u_1 \langle \mathcal{Z}_{u_n} \cdots \mathcal{Z}_{u_1} \rangle^e
$$
$$
= \sum_{n=0}^{\infty} (-i\lambda)^n \int_{t_{j-1}}^{t_j} \mathrm{d}u_n \cdots \int_{t_{j-1}}^{u_2} \mathrm{d}u_1 \Big( (\overline{\mathcal{Z}_{u_n} \mathcal{Z}_{u_{n-1}}} \cdots \overline{\mathcal{Z}_{u_4} \mathcal{Z}_{u_3}}\, \overline{\mathcal{Z}_{u_2} \mathcal{Z}_{u_1}})
$$
$$
+ (\overline{\mathcal{Z}_{u_n} \mathcal{Z}_{u_{n-1}}} \cdots \overline{\mathcal{Z}_{u_4} \overline{\mathcal{Z}_{u_3} \mathcal{Z}_{u_2}}\, \mathcal{Z}_{u_1}}) + \cdots
$$
$$
+ \langle\!\langle \mathcal{Z}_{u_n} \cdots \mathcal{Z}_{u_1} \rangle\!\rangle \Big).
$$

Note that the first-order cumulants are not present because $\langle \mathcal{Z}_u \rangle^e \propto \langle E(u^+) \rangle_{\rho^e} = 0$.

Therefore, the propagators can also be approximated with a perturbation theory based on bi-trajectory correlations. However, since there are no interventions dividing dynamical variables into blocks that could be then connected with increasingly weaker correlations, the theory appropriate for propagators should follow a principle other than the correlation strength hierarchy. On the other hand, regardless the form of the perturbation theory we will eventually

construct, it must be consistent with the hierarchy-based theory we are developing for the MTCs.

Previously, we have argued that the hierarchy-based perturbation theory is valid in the weak-coupling regime, in the sense that the higher-order corrections can be considered as a small perturbation to the zeroth-order QRF only if $\lambda \tau^e \ll 1$. Then, to make both perturbation theories consistence, we should focus on the propagator theory which leverages the weakness of system–environment coupling and the shortness of the correlation time. In particular, we expect from our theory to recover, in the lowest nontrivial order, the standard Born–Markov approximation:

$$\Lambda_{t_j, t_{j-1}} := \left\langle \mathrm{T} \, \mathrm{e}^{-\mathrm{i}\lambda \int_{t_{j-1}}^{t_j} \mathcal{Z}_u \mathrm{d}u} \right\rangle^e \approx \mathrm{e}^{(t_j - t_{j-1})\mathcal{L}^s}, \tag{69}$$

with the Davis generator $\mathcal{L}^s$ induced by the system–environment Hamiltonian $\hat{H}^{se}$:

$$\begin{aligned}
\mathcal{L}^s = &-\mathrm{i} \sum_\omega \sum_{\alpha,\alpha'} h^e_{\alpha,\alpha'}(\omega) \big[ \hat{V}^s_{\alpha'}(-\omega) \hat{V}^s_\alpha(\omega), \bullet \big] \\
&+ \sum_\omega \sum_{\alpha,\alpha'} w^e_{\alpha,\alpha'}(\omega) \Big( \hat{V}^s_\alpha(\omega) \bullet \hat{V}^s_{\alpha'}(-\omega) - \frac{1}{2} \{ \hat{V}^s_{\alpha'}(-\omega) \hat{V}^s_\alpha(\omega), \bullet \} \Big),
\end{aligned} \tag{70}$$

the jump operators,

$$\hat{V}^s_\alpha(\omega) = \sum_{k,k'} \delta\big(H^s(k) - H^s(k') - \omega\big) |k\rangle\langle k|\hat{V}^s_\alpha|k'\rangle\langle k'|, \tag{71}$$

defined with respect to frequencies of the renormalized system Hamiltonian (cf. Eq. (13)),

$$H^s, |k\rangle : \quad \left( \hat{H}^s_0 + \lambda \sum_\alpha \mathrm{tr}(\hat{V}^e_\alpha \hat{\rho}^e_0) \hat{V}^s_\alpha - H^s(k)\hat{1}^s \right) |k\rangle = 0, \tag{72}$$

and the transition rates, and energy level shifts given by the environmental two-time MTCs,

$$w^e_{\alpha,\alpha'}(\omega) = 2\lambda^2 \mathrm{Re} \int_0^\infty \langle E_\alpha(u^+) E_{\alpha'}(0^+) \rangle_{\rho^e} \, \mathrm{e}^{-\mathrm{i}\omega u} \mathrm{d}u, \tag{73}$$

$$h^e_{\alpha,\alpha'}(\omega) = -2\lambda^2 \mathrm{Im} \int_0^\infty \langle E_\alpha(u^+) E_{\alpha'}(0^+) \rangle_{\rho^e} \, \mathrm{e}^{-\mathrm{i}\omega u} \mathrm{d}u. \tag{74}$$

To achieve these goals, we shall base the propagator perturbation theory on the *super-cumulant series* expansion [12] involving the super-operator generators defined implicitly through the following equation:

$$\textbf{Super-cumulant series} : \quad \Lambda_{t,t'} \equiv \mathrm{T} \, \mathrm{e}^{\sum_{k=1}^\infty \lambda^k \int_{t'}^t \mathcal{L}^{(k)}_{u,t'} \mathrm{d}u} = \sum_{n=0}^\infty \int_{t'}^t \mathrm{d}u_n \cdots \int_{t'}^{u_2} \mathrm{d}u_1 \prod_{j=n}^1 \left( \sum_{k_j=1}^\infty \lambda^{k_j} \mathcal{L}^{(k_j)}_{u_j,t'} \right). \tag{75}$$

The explicit forms of super-cumulants $\mathcal{L}^{(k)}_{u,t'}$ constituting the series are found by expanding both sides of the equation into power series and comparing terms of corresponding order in parameter $\lambda$; in particular, the second super-cumulant reads:

$$\lambda^2 \mathcal{L}^{(2)}_{u,t'} = -\lambda^2 \int_{t'}^u \mathrm{d}u' \, \overline{\mathcal{Z}_u \mathcal{Z}_{u'}}. \tag{76}$$

(The first super-cumulant is zero, $\mathcal{L}^{(1)} \propto \langle \mathcal{Z} \rangle^e = 0$.)

The simplest form of propagator perturbation theory employing super-cumulant expansion consists of curtailing the series to finite number of terms:

$$
\textbf{$k_0^{\text{th}}$-order approximation} \qquad \sum_{k=2}^{\infty} \lambda^k \int_{t'}^{t} \mathcal{L}_{u,t'}^{(k)} \mathrm{d}u \approx \sum_{k=2}^{k_0} \lambda^k \int_{t'}^{t} \mathcal{L}_{u,t'}^{(k)} \mathrm{d}u. \tag{77}
$$

This coincides with the $(k_0 - 1)$th-order weak-coupling approximation because the $k$th super-cumulant scales roughly as

$$
\left\| \lambda^k \int_{t'}^{t} \mathcal{L}_{u,t'}^{(k)} \mathrm{d}u \right\| \sim \lambda(t-t')(\lambda \tau^e)^{k-1}. \tag{78}
$$

The principal advantage of using this approach is that, by construction, the obtained approximated propagators are *computable* maps: when one sets

$$
\Lambda_{t,t'} \approx \Lambda_{t,t'}^{(k_0)} \equiv \mathrm{T}\, e^{\sum_{k=2}^{k_0} \lambda^k \int_{t'}^{t} \mathcal{L}_{u,t'}^{(k)} \mathrm{d}u}, \tag{79}
$$

so that the map's generator is a well-defined, finite super-operator, one can then compute the propagator by integrating the corresponding dynamical equation [12]

$$
\frac{\mathrm{d}}{\mathrm{d}t} \Lambda_{t,t'}^{(k_0)} = \Big( \sum_{k=2}^{k_0} \lambda^k \mathcal{L}_{t,t'}^{(k)} \Big) \Lambda_{t,t'}^{(k_0)}, \qquad \Lambda_{t',t'}^{(k_0)} = \bullet, \tag{80}
$$

where the super-operator identity is expressed using the "bullet" notation, $\bullet \hat{A} = \hat{A}$.

Adopting the super-cumulant strategy, we immediately identify that the cut-off at $k_0 = 2$ results in the standard Born approximation:

$$
\begin{array}{c} \textbf{Born} \\ \textbf{approximation} \\ \textbf{($2^{\text{nd}}$-order)} \end{array} \qquad \Lambda_{t_j,t_{j-1}} \approx \Lambda_{t_j,t_{j-1}}^{(2)} : \quad \frac{\mathrm{d}}{\mathrm{d}t} \Lambda_{t,t_{j-1}}^{(2)} = -\lambda^2 \int_{t_{j-1}}^{t} \overbracket{\mathcal{Z}_t \mathcal{Z}_u}\, \mathrm{d}u\, \Lambda_{t,t_{j-1}}^{(2)}, \quad \Lambda_{t_{j-1},t_{j-1}}^{(2)} = \bullet.
$$
$$\tag{81}$$

The dynamical equation (81) can be further simplified within the assumed parameter regime: since the cumulant range is finite,

$$
\overbracket{\mathcal{Z}_u \mathcal{Z}_{u'}} \xrightarrow{u-u'>\tau^e} 0, \tag{82}
$$

and the correlation time is short, $t_j - t_{j-1} \gg \tau^e$, the limit of the integral in Eq. (76) can be extended to infinity, resulting in

**Markov approximation** :

$$
\begin{aligned}
-\lambda^2 \int_{t_{j-1}}^{t_j} \mathrm{d}u \int_{t_{j-1}}^{u} \mathrm{d}u' \overbracket{\mathcal{Z}_u \mathcal{Z}_{u'}} \approx \sum_{\omega,\omega'} \sum_{\alpha,\alpha'} \int_{t_{j-1}}^{t_j} e^{\mathrm{i}(\omega-\omega')u} \mathrm{d}u \Big[ & \big(\gamma_{\alpha,\alpha'}(\omega) + \gamma_{\alpha,\alpha'}(\omega')^*\big) \hat{V}_\alpha^s(\omega) \bullet \hat{V}_{\alpha'}^s(-\omega') \\
& -\gamma_{\alpha,\alpha'}(\omega) \hat{V}_{\alpha'}^s(-\omega') \hat{V}_\alpha^s(\omega) \bullet - \bullet \hat{V}_{\alpha'}^s(-\omega') \\
& \times \hat{V}_\alpha^s(\omega) \gamma_{\alpha,\alpha'}(\omega')^* \Big],
\end{aligned} \tag{83}
$$

where

$$
\gamma_{\alpha,\alpha'}(\omega) = \frac{w_{\alpha,\alpha'}^e(\omega) + \mathrm{i}h_{\alpha,\alpha'}^e(\omega)}{2} = \int_0^\infty \langle E_\alpha(u^+) E_{\alpha'}(0^+) \rangle_{\rho^e}\, e^{-\mathrm{i}\omega u} \mathrm{d}u. \tag{84}
$$

If we were to substitute this form of generator to the dynamical equation (81), we would obtain what amounts to the famous Redfield master equation for the propagator [9–11, 33].

Finally, if, in addition, we also assume that $t_j - t_{j-1}$ is long in comparison to the slowest time scale of the system $s$,

$$t_j - t_{j-1} \gg 2\pi / \min\left\{|\omega - \omega'| \mid \omega, \omega' = H^s(k) - H^s(k'), \omega \neq \omega'\right\}, \quad (85)$$

then, in Eq. (83), the 'off-resonance' $\omega \neq \omega'$ phase factors become much smaller than the 'on-resonance' $\omega = \omega'$ terms,

$$\left|\int_{t_{j-1}}^{t_j} e^{i(\omega - \omega')u} du\right| \ll \left|\int_{t_{j-1}}^{t_j} e^{i0u} du\right|. \quad (86)$$

In that case, we can apply the *secular* (or rotating-wave) approximation where the contributions from off-resonance terms is wholly neglected, leading at last to the Davis generator (70) we aimed to obtain:

$$\begin{array}{r}\textbf{Markov \& Secular}\\ \textbf{approximations}\end{array} : \quad -\lambda^2 \int_{t_{j-1}}^{t_j} du \int_{t_{j-1}}^{u} du' \overbrace{\mathcal{Z}_u \mathcal{Z}_{u'}} \approx (t_j - t_{j-1})\mathcal{L}^s. \quad (87)$$

## 6 Beyond quantum regression: The first-order perturbation

Given the general form of the open system bi-probability distribution (56),

$$Q_{\mathbf{t}_n|0}^{\mathbf{F}_n^s|\rho^{se}}(\mathbf{f}_n^+, \mathbf{f}_n^-) = \text{tr}_s\left[\left\langle \prod_{j=n}^{1} \mathcal{P}_j \, \mathrm{T} e^{-i\int_{t_{j-1}}^{t_j} \mathcal{Z}_u du}\right\rangle^e \hat{\rho}_0^s\right], \quad (88)$$

we now proceed to find the first-order correction to the zeroth-order quantum regression formula (QRF) approximation (66),

$$\textbf{0}^{\text{th}}\textbf{-order (QRF)} : \quad \prod_{j=n}^{1} \mathcal{P}_j \left\langle \mathrm{T} e^{-i\lambda \int_{t_{j-1}}^{t_j} \mathcal{Z}_u du}\right\rangle^e = \prod_{j=n}^{1} \mathcal{P}_j \Lambda_{t_j, t_{j-1}} \approx \prod_{j=n}^{1} \mathcal{P}_j e^{(t_j - t_{j-1})\mathcal{L}^s}. \quad (89)$$

First, we must identify specific terms that contribute to the perturbation: We are looking for two-time cumulants from the top of the correlation strength hierarchy that connect dynamical variables across interventions. However, since we assume $t_j - t_{j-1} \gg \tau^e$, due to finite range of cumulants, we can neglect connections between variables separate by more than one intervention. Bi-trajectory correlations connecting dynamical variables within propagators bracketed by interventions are already accounted for in a consistent manner by the Born–Markov approximation discussed previously. Applying these criteria we obtain the following decomposition:

$$\left\langle \prod_{j=n}^{1} \mathcal{P}_j \, \mathrm{T} e^{-i\lambda \int_{t_{j-1}}^{t_j} \mathcal{Z}_u du}\right\rangle^e - \prod_{j=n}^{1} \mathcal{P}_j \Lambda_{t_j, t_{j-1}} \approx \sum_{j=1}^{n-1} \left(\mathcal{P}_n \prod_{k=n-1}^{j+1} \Lambda_{t_{k+1}, t_k} \mathcal{P}_k\right) \delta\mathcal{P}_j \left(\prod_{k=j-1}^{1} \mathcal{P}_k \Lambda_{t_k, t_{k-1}}\right), \quad (90)$$

where

$$\delta\mathcal{P}_j = \left\langle \mathrm{T} e^{-i\lambda \int_{t_j}^{t_{j+1}} \mathcal{Z}_u du} \mathcal{P}_j \mathrm{T} e^{-i\lambda \int_{t_{j-1}}^{t_j} \mathcal{Z}_u du}\right\rangle^e - \left\langle \mathrm{T} e^{-i\lambda \int_{t_j}^{t_{j+1}} \mathcal{Z}_u du}\right\rangle^e \mathcal{P}_j \left\langle \mathrm{T} e^{-i\lambda \int_{t_{j-1}}^{t_j} \mathcal{Z}_u du}\right\rangle^e \quad (91)$$

$$\approx \sum_{k,\ell=0}^{\infty} (-i\lambda)^{k+\ell} \int_{t_j}^{t_{j+1}} du_k \cdots \int_{t_r}^{u_2} du_1 \int_{t_{j-1}}^{t_j} dv_\ell \cdots \int_{t_r}^{v_2} dv_1 \sum_{r=1}^{k} \sum_{r'=1}^{\ell} \left\langle \mathcal{Z}_{u_k} \cdots \overbrace{\mathcal{Z}_{u_r} \cdots \mathcal{Z}_{u_1}}^e \mathcal{P}_j \left\langle \mathcal{Z}_{v_\ell} \cdots \overline{\mathcal{Z}_{v_{r'}} \cdots \mathcal{Z}_{v_1}}\right\rangle^e\right.$$

$$= -\lambda^2 \int_{t_j}^{t_{j+1}} dt_> \int_{t_{j-1}}^{t_j} dt_< \left\langle \mathrm{T} e^{-i\lambda \int_{t_>}^{t_{j+1}} \mathcal{Z}_u du} \overline{\mathcal{Z}_{t_>} \mathrm{T} e^{-i\lambda \int_{t_j}^{t_>} \mathcal{Z}_u du}}\right\rangle^e \mathcal{P}_j \left\langle \mathrm{T} e^{-i\lambda \int_{t_<}^{t_j} \mathcal{Z}_u du} \mathcal{Z}_{t_<} \mathrm{T} e^{-i\int_{t_{j-1}}^{t_<} \mathcal{Z}_u du}\right\rangle^e.$$

This perturbation can be further simplified in the weak-coupling regime when we take note that, due to finite range of the cumulant

$$\overbrace{\mathcal{Z}_{t_>}\cdots\mathcal{Z}_{t_<}}^{} \xrightarrow{t_>-t_<>\tau^e} 0\,, \tag{92}$$

the integrals over $t_>$ and $t_<$ have natural cutoffs:

$$t_j+\tau^e \gtrsim t_> \geq t_j\,, \qquad t_j \geq t_< \gtrsim t_j-\tau^e\,,$$

which allows us to apply the following approximations:

1. The integration limits can be extended to infinity,

$$\int_{t_j}^{t_{j+1}} dt_> \int_{t_{j-1}}^{t_j} dt_< \overbrace{\mathcal{Z}_{t_>}\cdots\mathcal{Z}_{t_<}}^{} \approx \int_{t_j}^{\infty} dt_> \int_{-\infty}^{t_j} dt_< \overbrace{\mathcal{Z}_{t_>}\cdots\mathcal{Z}_{t_<}}^{}\,.$$

2. Since $t_>-t_< < \tau^e$, the propagators acting between variable $\mathcal{Z}_{t_>/t_<}$ and intervention $\mathcal{P}_j$, must have negligible effect,

$$\mathrm{T}\,e^{-i\lambda\int_{t_j}^{t_>}\mathcal{Z}_u du} \sim \mathrm{T}\,e^{-i\lambda\int_{t_j}^{t_j+\tau^e}\mathcal{Z}_u du} \sim e^{-i\lambda\tau_e\mathcal{Z}_{t_j}} \approx \bullet\,,$$

$$\mathrm{T}\,e^{-i\lambda\int_{t_<}^{t_j}\mathcal{Z}_u du} \sim \mathrm{T}\,e^{-i\lambda\int_{t_j-\tau^e}^{t_j}\mathcal{Z}_u du} \sim e^{-i\lambda\tau_e\mathcal{Z}_{t_j}} \approx \bullet\,.$$

3. For the same reason, the propagators surrounding the cumulant can be extended towards the mid point $t_j$,

$$\mathrm{T}\,e^{-i\lambda\int_{t_>}^{t_{j+1}}\mathcal{Z}_u du} \sim \mathrm{T}\,e^{-i\lambda\int_{t_j}^{t_{j+1}}\mathcal{Z}_u du}\,\mathrm{T}\,e^{-i\lambda\int_{t_j+\tau^e}^{t_j}\mathcal{Z}_u du} \approx \mathrm{T}\,e^{-i\lambda\int_{t_j}^{t_{j+1}}\mathcal{Z}_u du}\,,$$

$$\mathrm{T}\,e^{-i\lambda\int_{t_{j-1}}^{t_<}\mathcal{Z}_u du} \sim \mathrm{T}\,e^{-i\lambda\int_{t_j}^{t_j-\tau^e}\mathcal{Z}_u du}\,\mathrm{T}\,e^{-i\lambda\int_{t_{j-1}}^{t_j}\mathcal{Z}_u du} \approx \mathrm{T}\,e^{-i\lambda\int_{t_{j-1}}^{t_j}\mathcal{Z}_u du}\,.$$

These result in the following form of the first-order perturbation:

$$\mathbf{1^{st}\text{-order}}: \quad \sum_{j=1}^{n-1}\Big(\mathcal{P}_n\prod_{k=n-1}^{j+1}\Lambda_{t_{k+1},t_k}\mathcal{P}_k\Big)\Lambda_{t_{j+1},t_j}\Big(-\lambda^2\int_{t_j}^{\infty} du \int_{-\infty}^{t_j} du' \overbrace{\mathcal{Z}_u\mathcal{P}_j\mathcal{Z}_{u'}}^{}\Big)\Lambda_{t_j,t_{j-1}}\Big(\prod_{k=j-1}^{1}\mathcal{P}_k\Lambda_{t_k,t_{k-1}}\Big)\,. \tag{93}$$

Recalling that we are assuming stationary distribution for $e$, so that

$$\big\langle E_{\alpha_2}(u_2^{\pm})E_{\alpha_1}(u_1^{\pm})\big\rangle_{\rho^e} = \big\langle E_{\alpha_2}((u_2-t_0)^{\pm})E_{\alpha_1}((u_1-t_0)^{\pm})\big\rangle_{\rho^e}\,, \quad \text{for any } t_0\,, \tag{94}$$

the cumulant can be explicitly evaluated to the following form:

$$-\lambda^2\int_{t_j}^{\infty} du \int_{-\infty}^{t_j} du' \overbrace{\mathcal{Z}_u\mathcal{P}_j\mathcal{Z}_{u'}}^{} = -\lambda^2\int_{t_j}^{\infty} du \int_{-\infty}^{t_j} du' \overbrace{\mathcal{Z}_u\big(e^{i\hat{H}^s t_j}\bullet e^{-i\hat{H}^s t_j}\big)\big(\hat{P}^{F_j^s}(f_j^+)\bullet\hat{P}^{F_j^s}(f_j^-)\big)\big(e^{-i\hat{H}^s t_j}\bullet e^{i\hat{H}^s t_j}\big)\mathcal{Z}_{u'}}^{}$$

$$= \big(e^{i\hat{H}^s t_j}\bullet e^{-i\hat{H}^s t_j}\big)\times\sum_{\alpha,\omega}[\hat{V}_\alpha^s(\omega),\bullet]\big(\hat{P}^{F_j^s}(f_j^+)\bullet\hat{P}^{F_j^s}(f_j^-)\big)$$

$$\times\sum_{\alpha',\omega'}\Big(C_{\alpha,\alpha'}(\omega,\omega')[\hat{V}_{\alpha'}^s(-\omega'),\bullet]+iK_{\alpha,\alpha'}(\omega,\omega')\{\hat{V}_{\alpha'}^s(-\omega'),\bullet\}\Big)$$

$$\times\big(e^{-i\hat{H}^s t_j}\bullet e^{i\hat{H}^s t_j}\big)\,, \tag{95}$$

where the information about the temporal correlations in $e$ is confined to coefficients

$$C_{\alpha,\alpha'}(\omega,\omega') = -\lambda^2\int_0^{\infty} du\,e^{i(\omega-\omega')u}\int_u^{\infty} dv\,e^{i\omega' v}\,\mathrm{Re}\langle E_\alpha(v^+)E_{\alpha'}(0^+)\rangle_{\rho^e}\,, \tag{96}$$

$$K_{\alpha,\alpha'}(\omega,\omega') = -\lambda^2\int_0^{\infty} du\,e^{i(\omega-\omega')u}\int_u^{\infty} dv\,e^{i\omega' v}\,\mathrm{Im}\langle E_\alpha(v^+)E_{\alpha'}(0^+)\rangle_{\rho^e}\,. \tag{97}$$

# 7 Application to the problem of thermalization propagation

We now come back to the problem of thermalization propagation. We will show here how the developed MTC perturbation theory applies in this example, and how exactly it leads to the first-order correction which recover the detailed balance necessary for further 'spread' of thermalization.

The two-time MTC (17) required to calculate the transition rates $w^q(\omega)$ can now be expressed as the second moment of the corresponding bi-probability distribution,

$$
\begin{aligned}
\langle \sigma_z^q(t_2^+)\sigma_z^q(t_1^+)\rangle_{p_\beta^q} &= \sum_{\boldsymbol{\sigma}_2^\pm \in \{1,-1\}^2} \sigma_2^+ \sigma_1^+ Q_{\mathbf{t}_2|0}^{\sigma_z^q|p_\beta^q}(\boldsymbol{\sigma}_2^+,\boldsymbol{\sigma}_2^-) \\
&= \sum_{\boldsymbol{\sigma}_2^\pm \in \{1,-1\}^2} \sigma_2^+ \sigma_1^+ \,\mathrm{tr}\left[\left\langle \prod_{j=2}^1 \mathcal{P}_j^q \,\mathrm{T}\, e^{-i\lambda \int_{t_{j-1}}^{t_j} \mathcal{Z}_u^q du}\right\rangle e^{} \hat{p}_\beta^q \right],
\end{aligned}
\tag{98}
$$

where, given that $\hat{H}^q = 0$ and $\mathrm{tr}_b(\hat{V}^b \hat{p}_\beta^b) = 0$, so that $\hat{V}^b = \hat{E}$, the intervention and the dynamical variable have simple forms:

$$
\mathcal{P}_j^q = |\sigma_j^+\rangle\langle\sigma_j^+| \bullet |\sigma_j^-\rangle\langle\sigma_j^-|, \qquad \mathcal{Z}_u^q = \sum_{\varepsilon^\pm} \left(e(\varepsilon^+)\hat{\sigma}_x^q \bullet - \bullet \hat{\sigma}_x^q e(\varepsilon^-)\right)\delta_{\varepsilon^+,\varepsilon^+(u,\vec{\varphi})}\delta_{\varepsilon^-,\varepsilon^-(u,\vec{\varphi})}, \tag{99}
$$

with $(\hat{\sigma}_z^q - \sigma\hat{1}^q)|\sigma\rangle = 0$ and $(\hat{V}^b - e(\varepsilon)\hat{1}^b)\exp(-i\vec{\varphi}\cdot\vec{\hat{T}}^b)|\varepsilon\rangle = 0$.

At this point we can simply use the derived formulas (89) and (93) to calculate the bi-probability distribution—and then the MTC—up to the first-order of our perturbation theory; thus we have the zeroth-order QRF contribution:

$$
\begin{aligned}
\left(\prod_{j=2}^1 \mathcal{P}_j^q \Lambda_{t_j,t_{j-1}}^q\right)\hat{p}_\beta^q &= |\sigma_2^+\rangle\langle\sigma_2^-|\,\langle\sigma_2^+|\left(e^{(t_2-t_1)\mathcal{L}^q}|\sigma_1^+\rangle\langle\sigma_1^-|\right)|\sigma_2^-\rangle \cdot \langle\sigma_1^+|\left(\tfrac{1}{2}e^{t_1\mathcal{L}^q}\hat{1}^q\right)|\sigma_1^-\rangle \\
&= \delta_{\sigma_2^+,\sigma_2^-}\delta_{\sigma_1^+,\sigma_1^-}\frac{1+(\sigma_2^+\cdot\sigma_1^+)e^{-2w^b(0)(t_2-t_1)}}{4}|\sigma_2^+\rangle\langle\sigma_2^+|,
\end{aligned}
\tag{100}
$$

and the first-order correction:

$$
\begin{aligned}
-\lambda^2 \mathcal{P}_2^q \Lambda_{t_2,t_1}^q &\left(\int_{t_1}^\infty du \int_{-\infty}^{t_1} du' \overbrace{\mathcal{Z}_u^q \mathcal{P}_1^q \mathcal{Z}_{u'}^q}\right)\Lambda_{t_1,0}^q \hat{p}_\beta^q \\
&= |\sigma_2^+\rangle\langle\sigma_2^-| \cdot \langle\sigma_2^+|\left(e^{(t_2-t_1)\mathcal{L}^q}\left[\hat{\sigma}_x^q,|\sigma_1^+\rangle\langle\sigma_1^-|\right]\right)|\sigma_2^-\rangle \\
&\quad \times \langle\sigma_1^+|\tfrac{1}{2}\left(C(0,0)[\hat{\sigma}_x^q,\bullet]+iK(0,0)\{\hat{\sigma}_x^q,\bullet\}\right)e^{t_1\mathcal{L}^q}\hat{1}^q|\sigma_1^-\rangle \\
&= -i\delta_{\sigma_2^+,\sigma_2^-}\delta_{\sigma_1^-,-\sigma_1^+}(\sigma_2^+\cdot\sigma_1^+)K(0,0)e^{-2w^b(0)(t_2-t_1)}|\sigma_2^+\rangle\langle\sigma_2^+|.
\end{aligned}
\tag{101}
$$

We immediately confirm that the zeroth-order contribution (100) indeed reproduced the quantum regression formula approximation that lead to the inaccurate prediction of unit transition rates ratio (19):

$$
\mathbf{QRF:} \quad \sum_{\boldsymbol{\sigma}_2^\pm} \sigma_2^+ \sigma_1^+ \,\mathrm{tr}\left[\left(\prod_{j=2}^1 \mathcal{P}_j^q \Lambda_{t_j,t_{j-1}}^q\right)\hat{p}_\beta^q\right] = e^{-2w^b(0)(t_2-t_1)}.
$$

To calculate the first-order correction, and see if we can obtain a more accurate transition rates in the process, we require an explicit value of the coefficient $K(0,0)$; as per definition (97), this forces us to decide on a concrete model of bath correlations. For simplicity,

we shall assume an exponential auto-correlation function (cf. Eq (38a)) that gives us a well-defined correlation time $\tau^b$:

$$C_{u_2,u_1}^{V^b} = \text{Re}\langle E(u_2^+)E(u_1^+)\rangle_{p_\beta^b} \equiv \exp\left(-\frac{|u_2-u_1|}{\tau^b}\right).$$
(102)

Then, the imaginary part of the MTC—which is required by $K(0,0)$—can be obtained from the fluctuation–dissipation theorem (39); even with exponential correlation function an analytical result is only possible in high bath temperature regime, $\beta \ll \tau^b$, which we tacitly assume here (cf. appendix B):

$$\text{Im}\langle E(u^+)E(0^+)\rangle_{p_\beta^b} = \int_{-\infty}^{\infty} i\tanh\left(\frac{\beta\omega}{2}\right)\left(\int_{-\infty}^{\infty}\text{Re}\langle E(v^+)E(0^+)\rangle_{p_\beta^b}e^{-i\omega v}dv\right)e^{i\omega u}\frac{d\omega}{2\pi}$$
$$\xrightarrow{u\gg\tau^b,\,\beta\ll\tau^b} -\frac{\beta}{2\tau^b}\exp\left(-\frac{u}{\tau^b}\right).$$
(103)

The resultant bi-probability distribution reads:

$$Q_{t_2|0}^{\sigma_z^q|p_\beta^q}(\sigma_2^+,\sigma_2^-) \approx \text{tr}\left[\left(\prod_{j=2}^{1}\mathcal{P}_j^q\Lambda_{t_j,t_{j-1}}^q - \lambda^2\mathcal{P}_2^q\Lambda_{t_2,t_1}^q\int_{t_1}^{\infty}du\int_{-\infty}^{t_1}du'\,\overline{\mathcal{Z}_u^q\mathcal{P}_1^q\mathcal{Z}_{u'}^q}\Lambda_{t_1,0}^q\right)\hat{p}_\beta^q\right]$$
$$= \delta_{\sigma_2^+,\sigma_2^-}\delta_{\sigma_1^+,\sigma_1^-}\frac{1+(\sigma_2^+\cdot\sigma_1^+)e^{-2w^b(0)(t_2-t_1)}}{4}$$
$$-\frac{i}{2}\lambda^2\tau^b\beta\,\delta_{\sigma_2^+,\sigma_2^-}\delta_{\sigma_1^-,-\sigma_1^+}(\sigma_2^+\cdot\sigma_1^+)e^{-2w^b(0)(t_2-t_1)},$$
(104)

which then gives us the following transition rate:

$$w^q(\omega) = 2\mu^2\text{Re}\int_0^{\infty}\langle\sigma_z^q(u^+)\sigma_z^q(0^+)\rangle_{p_\beta^q}e^{-i\omega u}du = 2\mu^2\left(\frac{2w^b(0)-4\lambda^2\tau^b\beta\omega}{(2w^b(0))^2+\omega^2}\right)$$
$$= \frac{8\mu^2\lambda^2\tau^b}{(4\lambda^2\tau^b)^2+\omega^2}\left(1-\frac{\beta\omega}{2}\right),$$
(105)

as $w^b(0) = 2\lambda^2\tau^b$ for the chosen exponential auto-correlation (102). Therefore, the first-order perturbation brings about the crucial information about the bath temperature, and it leads to approximated detailed balance (21) we described previously.

## 8 Conclusions

We have developed a perturbation theory for computing multi-time correlation functions in open quantum systems. The theory is based on a hierarchy of temporal correlation strengths, which naturally emerges in open system problems when described within the recently introduced bi-trajectory formalism for quantum mechanics.

We demonstrated that the quantum regression formula (QRF) method—a commonly used *ad hoc* approximation—is equivalent to the zeroth-order perturbation in our theory. However, we also showed the inadequacy of QRF, as it fails to account for the propagation of thermalization, underscoring the need for a systematic perturbation theory capable of capturing important physical effects through higher-order corrections. Indeed, in the case of the qubit–bath system studied in our example, inclusion of the lowest-order perturbation was sufficient to reproduce qualitatively correct behavior.

Additionally, we introduced the super-cumulant expansion method for approximating dynamical maps and propagators in general. The lowest-order approximation—applicable in the

weak-coupling regime—yields results identical to those of the conventional Born–Markov approximation. However, our approach does not rely on the problematic Born–Markov ansatz, nor on any other assumptions regarding the state of the total system.

Previous works—e.g., [34–37]—have proposed approximation schemes to improve on the QRF method. The approach adopted in these works is, in a sense, complementary to ours. The standard strategy is to address the problem of MTCs by extending the so-called quantum regression theorem (QRT) beyond the zero-correlation-time regime. Essentially, the QRT states that when $\tau^e \to 0$, the single- and two-time correlations of system observables satisfy the same dynamical equation [37]; hence, QRT implies the exactness of QRF for zero-range cumulants, which is, of course, consistent with our theory. "Extending QRT" then amounts to writing down the derivative of a two-time MTC with respect to the second time argument and proposing an approximation that turns the expression into a legitimate dynamical equation—that is, closing the equation by reassembling the MTC on the right-hand side. Such an approximation is necessary because, as per QRT, the equation does not close on its own when $\tau^e > 0$.

It follows that theories utilizing this dynamical-equation approach result in MTC approximations analogous in form to the QRF method, where there are no explicit connections reaching across interventions. Instead, the deployed approximations lead to effective propagators between interventions that attempt to capture the effects that would result from cross-intervention connections, were they actually included. In contrast, our approach facilitates such cross-intervention connections, thereby allowing for the independent treatment of the in-between propagators, as demonstrated in Sec. 5. Seemingly, there is no decisive argument favoring our approach over the traditional equation-of-motion method. When faced with a concrete problem, one must carefully consider the pros and cons of each method, and which approach is preferable will most likely depend on the fine details. However, there is something to be said for how intuitive it is to build up corrections to the MTC by progressively including more complex temporal correlations connecting the propagators across interventions—no such clear picture can be used to illustrate the physics of perturbation theory within the equation-of-motion framework.

Finally, we note that the hierarchy of correlation strength (49), which constitutes the basis for our theory, can be analyzed in terms of *graphs*. In this reading, one interpretation is to represent the moments of dynamical variables as the graph's vertices, and the edges connecting vertices as cumulants. Therefore, the introduced perturbation theory should naturally lend itself to some form of *diagrammatic* treatment. This opens a promising avenue for further development of the theory, with the prospect of a high payoff that is characteristic of diagrammatic methods known from other perturbation theories compatible with such an approach.

## Acknowledgments

The Author would like to extend their sincere thanks to P. Wysocki for extensive discussions that were instrumental for finalizing this manuscript.

## A Bi-probability distribution for open system

Following the setup from sec. 4, we consider here a composite system with the bipartite Hilbert space $\mathcal{H}^s \otimes \mathcal{H}^e$ and the Hamiltonian and initial state given by Eqs. (52) and (53), respectively.

The goal is to evaluate a bi-probability distribution associated with a sequence of *s*-only

observables $\mathbf{F}_n^s$, i.e., each $F_j^s$ is represented by an operator of form

$$\hat{F}_j^s \otimes \hat{1}^e, \qquad \hat{F}_j^s(t) = \sum_{f' \in \Omega(F_j)} f' \hat{P}_t^{F_j^s}(f') = \sum_{f' \in \Omega(F_j)} f' e^{i\hat{H}^s t} \hat{P}^{F_j^s} e^{-i\hat{H}^s t}. \tag{A.1}$$

According to the definition (29), the bi-probability is given by:

$$Q_{\mathbf{t}_n|0}^{\mathbf{F}_n^s|\rho^{se}}(\mathbf{f}_n^+, \mathbf{f}_n^-) = \mathrm{tr}\Bigg[\Bigg(\prod_{j=n}^{1} e^{i\hat{H}^{se} t_j} \hat{P}^{F_j^s}(f_j^+) \otimes \hat{1}^e e^{-i\hat{H}^{se} t_j}\Bigg)\hat{\rho}_0^s \otimes \hat{\rho}_0^e \Bigg(\prod_{j=1}^{n} e^{i\hat{H}^{se} t_j} \hat{P}_0^{F_j^s}(f_j^-) \otimes \hat{1}^e e^{-i\hat{H}^{se} t_j}\Bigg)\Bigg]$$

$$= \mathrm{tr}\Bigg[\prod_{j=n}^{1}\Big(\hat{P}_{t_j}^{F_j^s}(f_j^+) \otimes \hat{1}^e \, \hat{U}_{t_j, t_{j-1}}^{se}\Big)\hat{\rho}_0^s \otimes \hat{\rho}_0^e \prod_{j=1}^{n}\Big(\hat{U}_{t_{j-1}, t_j}^{se} \, \hat{P}_{t_j}^{F_j^s}(f_j^-) \otimes \hat{1}^e\Big)\Bigg], \tag{A.2}$$

where we have switched to the interaction picture,

$$\hat{U}_{t,t'}^{se} = \mathrm{T}\exp\Bigg(-i\lambda \int_{t'}^{t} \sum_\alpha \hat{V}_\alpha^s(u) \otimes \hat{E}_\alpha(u) \mathrm{d}u\Bigg), \qquad \hat{V}_\alpha^s(u) = e^{i\hat{H}^s u}\hat{V}_\alpha^s e^{-i\hat{H}^s u}, \quad \hat{E}_\alpha(u) = e^{i\hat{H}^e u}\hat{E}_\alpha e^{-i\hat{H}^e u}. \tag{A.3}$$

Then, we rearrange the operators under the trace into a composition of corresponding super-operators:

$$Q_{\mathbf{t}_n|0}^{\mathbf{F}_n^s|\rho^{se}}(\mathbf{f}_n^+, \mathbf{f}_n^-) = \mathrm{tr}\Bigg[\prod_{j=n}^{1}\Big(\big(\hat{P}_{t_j}^{F_j^s}(f_j^+) \otimes \hat{1}^e \bullet \hat{P}_{t_j}^{F_j^s}(f_j^-) \otimes \hat{1}^e\big)\big(\hat{U}_{t_j, t_{j-1}}^{se} \bullet \hat{U}_{t_{j-1}, t_j}^{se}\big)\Big)\hat{\rho}_0^s \otimes \hat{\rho}_0^e\Bigg],$$

so that we can invoke later down the line the following series expansion

$$\hat{U}_{t,t'}^{se} \bullet \hat{U}_{t',t}^{se} = \mathrm{T}\, e^{-i\int_{t'}^{t} \sum_\alpha \hat{V}_\alpha^s(u) \otimes \hat{E}_\alpha(u) \mathrm{d}u} \bullet \Big(\mathrm{T}\, e^{-i\int_{t'}^{t} \sum_\alpha \hat{V}_\alpha^s(u) \otimes \hat{E}_\alpha(u) \mathrm{d}u}\Big)^\dagger$$

$$= \mathrm{T}\exp\Bigg(-i\lambda \int_{t'}^{t} \mathrm{d}u \sum_\alpha \big[\hat{V}_\alpha^s(u) \otimes \hat{E}_\alpha(u), \bullet\big]\Bigg) \tag{A.4}$$

$$= \sum_{m=0}^{\infty}(-i\lambda)^m \int_{t'}^{t} \mathrm{d}u_m \sum_{\alpha_m}\big[\hat{V}_{\alpha_m}^s(u_m) \otimes \hat{E}_{\alpha_m}(u_m), \bullet\big]\cdots\int_{t'}^{u_2}\mathrm{d}u_1 \sum_{\alpha_1}\big[\hat{V}_{\alpha_1}^s(u_1) \otimes \hat{E}_{\alpha_1}(u_1), \bullet\big].$$

Using the spectral decomposition of the $e$-side coupling operators,

$$\hat{E}_\alpha(u) = \sum_\varepsilon e_\alpha(\varepsilon) e^{i\hat{H}^e u} e^{-i\vec{\varphi}_\alpha \cdot \vec{\hat{T}}^e}|\varepsilon\rangle\langle\varepsilon| e^{i\vec{\varphi}_\alpha \cdot \vec{\hat{T}}^e} e^{-i\hat{H}^e u} = \sum_{e' \in \Omega(E_\alpha)} e' \hat{P}_u^{E_\alpha}(e'), \tag{A.5}$$

with the reference basis $\{|\varepsilon\rangle \in \mathcal{H}^e \mid \varepsilon = 1, \ldots, d^e\}$ and the basis transformation parameterization $\exp(-i\vec{\varphi} \cdot \vec{\hat{T}}^e)$ defined as in section 2.2, and

$$\hat{P}_u^{E_\alpha}(e)\hat{P}_u^{E_\alpha}(e') = \delta_{e,e'}\hat{P}_u^{E_\alpha}(e), \qquad \sum_{e' \in \Omega(E_\alpha)} \hat{P}_u^{E_\alpha}(e') = \hat{1}^e, \tag{A.6}$$

we analyze the action of the generator,

$$\sum_\alpha \big[\hat{V}_\alpha^s(u) \otimes \hat{E}_\alpha(u), \bullet\big] = \sum_\alpha \Big(\hat{V}_\alpha^s(u) \otimes \hat{E}_\alpha(u) \bullet - \bullet \hat{V}_\alpha^s(u) \otimes \hat{E}_\alpha(u)\Big)$$

$$= \sum_\alpha \Big(\sum_{e^+ \in \Omega(E_\alpha)} e^+ \hat{V}_\alpha^s(u) \otimes \hat{P}_u^{E_\alpha}(e^+) \bullet \hat{1}^s \otimes \hat{1}^e - \hat{1}^s \otimes \hat{1}^e$$

$$\bullet \sum_{e^- \in \Omega(E_\alpha)} \hat{V}_\alpha^s(u) \otimes \hat{P}_u^{E_\alpha}(e^-) e^-\Big)$$

$$= \sum_\alpha \sum_{e^\pm \in \Omega(E_\alpha)}\Big(e^+ \hat{V}_\alpha^s(u) \otimes \hat{P}_u^{E_\alpha}(e^+) \bullet \hat{1}^s \otimes \hat{P}_u^{E_\alpha}(e^-) - \hat{1}^s \otimes \hat{P}_u^{E_\alpha}(e^+)$$

$$\bullet \hat{V}_\alpha^s(u) \otimes \hat{P}_u^{E_\alpha}(e^-) e^-\Big). \tag{A.7}$$

Substituting the expanded form of the evolution super-operators into the formula we obtain:

$$
\begin{aligned}
Q_{\mathbf{t}_n|0}^{F_n^s|\rho^s}(\mathbf{f}_n^+, \mathbf{f}_n^-) &= \prod_{i=1}^{n}\left( \sum_{m_i=0}^{\infty}(-i\lambda)^{m_i}\int_{t_{i-1}}^{t_i}du_{i;m_i}\cdots\int_{t_{i-1}}^{u_{i;2}}du_{i;1}\sum_{\alpha_{i;m_i}}\cdots\sum_{\alpha_{i;1}}\sum_{e_{i;m_i}^\pm}\cdots\sum_{e_{i;1}^\pm}\right)\\
&\quad\times \operatorname{tr}_s\left[\left(\prod_{j=n}^{1}\hat{P}_{t_j}^{F_j^s}(f_j^+)\bullet\hat{P}_{t_j}^{F_j^s}(f_j^-)\prod_{k_j=m_j}^{1}\left\{e_{j;k_j}^+\hat{V}_{\alpha_{j;k_j}}^s(u_{j;k_j})\bullet - \bullet\hat{V}_{\alpha_{j;k_j}}^s(u_{j;k_j})e_{j;k_j}^-\right\}\right)\hat{\rho}_0^s\right]\\
&\quad\times \operatorname{tr}_e\left[\hat{P}_{u_{n;m_n}}^{E_{\alpha_{n;m_n}}}(e_{n;m_n}^+)\cdots\hat{P}_{u_{1;1}}^{E_{\alpha_{1;1}}}(e_{1;1}^+)\hat{\rho}_0^e\hat{P}_{u_{1;1}}^{E_{\alpha_{1;1}}}(e_{1;1}^-)\cdots\hat{P}_{u_{n;m_n}}^{E_{\alpha_{n;m_n}}}(e_{n;m_n}^-)\right]\\
&= \prod_{i=1}^{n}\left( \sum_{m_i=0}^{\infty}(-i\lambda)^{m_i}\int_{t_{i-1}}^{t_i}du_{i;m_i}\cdots\int_{t_{i-1}}^{u_{i;2}}du_{i;1}\sum_{\alpha_{i;m_i}}\cdots\sum_{\alpha_{i;1}}\sum_{e_{i;m_i}^\pm}\cdots\sum_{e_{i;1}^\pm}\right)\\
&\quad\times \operatorname{tr}_s\left[\left(\prod_{j=n}^{1}\hat{P}_{t_j}^{F_j^s}(f_j^+)\bullet\hat{P}_{t_j}^{F_j^s}(f_j^-)\prod_{k_j=m_j}^{1}\left\{e_{j;k_j}^+\hat{V}_{\alpha_{j;k_j}}^s(u_{j;k_j})\bullet - \bullet\hat{V}_{\alpha_{j;k_j}}^s(u_{j;k_j})e_{j;k_j}^-\right\}\right)\hat{\rho}_0^s\right]\\
&\quad\times Q_{u_{n;m_n},\ldots,u_{1;1}|0}^{E_{\alpha_{n;m_n}},\ldots,E_{\alpha_{1;1}}|\rho^e}(e_{n;m_n}^+,\ldots,e_{1;1}^+;e_{n;m_n}^-,\ldots,e_{1;1}^-),
\end{aligned}
\tag{A.8}
$$

where we recognize in the decomposition bi-probability distributions, but this time, associated with the environmental coupling operators.

According to sec. 2.2, bi-probability distributions associated with sequences of observables $E_\alpha$ are the discrete-time restrictions of the environmental bi-trajectory distribution $Q^e[\varepsilon^+,\varepsilon^-]$ (see Eq. (34)):

$$
\begin{aligned}
&Q_{u_{n;m_n},\ldots,u_{1;1}|0}^{E_{\alpha_{n;m_n}},\ldots,E_{\alpha_{1;1}}|\rho^e}\left(e_{n;m_n}^+,\ldots,e_{1;1}^+;e_{n;m_n}^-,\ldots,e_{1;1}^-\right)\\
&=\iint\Delta_{u_{n;m_n},\ldots,u_{1;1}|0}^{E_{\alpha_{n;m_n}},\ldots,E_{\alpha_{1;1}}|\rho^e}(e_{n;m_n}^+,\ldots,e_{1;1}^+;e_{n;m_n}^-,\ldots,e_{1;1}^-\mid\varepsilon^+,\varepsilon^-)Q^e[\varepsilon^+,\varepsilon^-][D\varepsilon^+][D\varepsilon^-]\\
&=\iint\left(\prod_{i=1}^{n}\prod_{k_i=1}^{m_i}\delta_{e_{i;k_i}^+,e_{\alpha_{i;k_i}}(\varepsilon^+(u_{i;k_i},\vec{\varphi}_{\alpha_{i;k_i}}))}\delta_{e_{i;k_i}^-,e_{\alpha_{i;k_i}}(\varepsilon^-(u_{i;k_i},\vec{\varphi}_{\alpha_{i;k_i}}))}\right)\\
&\quad\times\left(\sum_{\varepsilon_0}p^e(\varepsilon_0)\delta_{\varepsilon_0,\varepsilon^+(0,\vec{\varphi}_0)}\delta_{\varepsilon_0,\varepsilon^-(0,\vec{\varphi}_0)}\right)Q^e[\varepsilon^+,\varepsilon^-][D\varepsilon^+][D\varepsilon^-],
\end{aligned}
\tag{A.9}
$$

where $\hat{\rho}_0^e=\sum_{\varepsilon_0}p^e(\varepsilon_0)e^{-i\vec{\varphi}_0\cdot\vec{T}^e}|\varepsilon_0\rangle\langle\varepsilon_0|e^{i\vec{\varphi}_0\cdot\vec{T}^e}$. We can now make use of the discrete-time filters to evaluate the sums over eigenvalues of couplings $E_\alpha$ to turn coefficients $e_{j;k_j}^\pm$ into the corresponding bi-trajectory-dependent dynamical variables:

$$
\sum_{e_{j;k_j}^\pm}e_{j;k_j}^\pm\delta_{e_{j;k_j}^\pm,e_{\alpha_{j;k_j}}(\varepsilon^\pm(u_{j;k_j},\vec{\varphi}_{j;k_j}))}=e_{\alpha_{j;k_j}}\left(\varepsilon^\pm(u_{j;k_j},\vec{\varphi}_{\alpha_{j;k_j}})\right).
\tag{A.10}
$$

Recalling the definition of the super-operator dynamical variable (57),

$$
\mathscr{Z}_u=\sum_\alpha\left(e_\alpha(\varepsilon^+(u,\vec{\varphi}_\alpha))\hat{V}_\alpha^s(u)\bullet - \bullet\hat{V}_\alpha^s(u)e_\alpha(\varepsilon^-(u,\vec{\varphi}_\alpha))\right),
\tag{A.11}
$$

we find that after the filters have been reduced, the open system bi-probability simplifies to

$$
Q_{\mathbf{t}_n|0}^{\mathbf{F}_n^s|\rho^{se}}(\mathbf{f}_n^+,\mathbf{f}_n^-) = \iint \mathrm{tr}_s\left[\left(\prod_{j=n}^1 \hat{P}_{t_j}^{F_j^s}(f_j^+)\bullet\hat{P}_{t_j}^{F_j^s}(f_j^-)\left\{\sum_{m=0}^\infty(-i\lambda)^m\int_{t_{j-1}}^{t_j}du_m\mathcal{Z}_{u_m}\cdots\int_{t_{j-1}}^{u_2}du_1\mathcal{Z}_{u_1}\right\}\right)\hat{\rho}_0^s\right]
$$

$$
\times\left(\sum_{\varepsilon_0}p^e(\varepsilon_0)\delta_{\varepsilon_0,\varepsilon^+(0,\vec{\varphi}_0)}\delta_{\varepsilon_0,\varepsilon^-(0,\vec{\varphi}_0)}\right)Q^e[\varepsilon^+,\varepsilon^-][D\varepsilon^+][D\varepsilon^-]
$$

$$
= \iint \mathrm{tr}_s\left[\left(\prod_{j=n}^1 \hat{P}_{t_j}^{F_j^s}(f_j^+)\bullet\hat{P}_{t_j}^{F_j^s}(f_j^-)\,\mathrm{T}\,e^{-i\lambda\int_{t_{j-1}}^{t_j}\mathcal{Z}_u du}\right)\hat{\rho}_0^s\right]
$$

$$
\times\left(\sum_{\varepsilon_0}p^e(\varepsilon_0)\delta_{\varepsilon_0,\varepsilon^+(0,\vec{\varphi}_0)}\delta_{\varepsilon_0,\varepsilon^-(0,\vec{\varphi}_0)}\right)Q^e[\varepsilon^+,\varepsilon^-][D\varepsilon^+][D\varepsilon^-]
$$

$$
= \mathrm{tr}_s\left[\left\langle\prod_{j=n}^1\mathcal{P}_j\,\mathrm{T}\,e^{-i\lambda\int_{t_{j-1}}^{t_j}\mathcal{Z}_u du}\right\rangle^e\hat{\rho}_0^s\right], \tag{A.12}
$$

where we used the definition of the bi-trajectory average (61) and intervention (63).

# B Susceptibility

The goal is to evaluate the Fourier transform for the susceptibility,

$$
I(t) := \mathrm{Im}\langle E(t^+)E(0^+)\rangle_{p_\beta^e} = \int_{-\infty}^\infty i\tanh\left(\frac{\beta\omega}{2}\right)S(\omega)e^{i\omega t}\frac{d\omega}{2\pi}, \tag{B.1}
$$

where $S(\omega)=\int_{-\infty}^\infty C_{t,0}^E e^{-i\omega t}dt$ is the spectral density.

We shall proceed by employing the residue method. For $t>0$ we can complete the contour with the semicircle in the upper half of the complex plain,

$$
I(t) - \lim_{r\to\infty}\int_0^\pi\frac{d\varphi}{2\pi}e^{ire^{i\varphi}}\frac{1-e^{-\beta re^{i\varphi}}}{1+e^{-\beta re^{i\varphi}}}S(re^{i\varphi}) = -\sum_{z_p}\mathrm{Res}_{z_p}\left[S(z)\tanh\left(\frac{\beta z}{2}\right)\right]e^{iz_p t}. \tag{B.2}
$$

The semi-circle integral vanishes and we shall assume that the spectral density has only simple poles, e.g., $S(\omega)=2\tau^e/(1+(\tau^e\omega)^2)$; then, the only non-trivial element are the infinite number of simple poles of tanh,

$$
I(t) = -\sum_{z_p}\mathrm{Res}_{z_p}[S(z)]\tanh\left(\frac{\beta z_p}{2}\right)e^{iz_p t} \tag{B.3}
$$

$$
-2\sum_{n=0}^\infty\mathrm{Res}_{z_n}\left[\frac{1}{1+e^{-\beta z}}\right]e^{-(2n+1)\frac{\pi}{\beta}t}S\left(i(2n+1)\frac{\pi}{\beta}\right)
$$

$$
= -\tan\left(\frac{\beta}{2\tau^e}\right)e^{-\frac{t}{\tau^e}}-\frac{4\tau^e}{\beta}\sum_{n=0}^\infty\frac{e^{-(2n+1)\frac{\pi}{\beta}t}}{1-(2n+1)^2\pi^2\frac{(\tau^e)^2}{\beta^2}}
$$

$$
= -\tan\left(\frac{\beta}{2\tau^e}\right)e^{-\frac{t}{\tau^e}}-\frac{1}{\pi}\left[e^{\frac{t}{\tau^e}}B_{e^{-2\pi t/\beta}}\left(\frac{1}{2}+\frac{\beta}{2\pi\tau^e},0\right)-e^{-\frac{t}{\tau^e}}B_{e^{-2\pi t/\beta}}\left(\frac{1}{2}-\frac{\beta}{2\pi\tau^e},0\right)\right],
$$

where we have assumed that $\beta/(2\pi\tau^e)$ is not an integer, and the Euler beta function is defined as

$$
B_z(a,b) = \int_0^z s^{a-1}(1-s)^{b-1}ds. \tag{B.4}
$$

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
