# Peer review of "A perturbation theory for multi-time correlation functions in open quantum systems"

_SciPost Physics, doi:SciPost Phys. 19, 066 (2025)_

## Round 2 · Referee Report · Anonymous (Referee 1) · 2025-5-28

Strengths

Paper is well-written, which is almost never true for open systems papers.

Weaknesses

While the Introduction explains the problem with great clarity, the abstract does not. It says little about what is gained by computing multi-time correlations.

It might help the reader to outline the structure of the paper at the end of the introduction.

Report

The introduction of the paper is well-written and identifies the central problem in a clear manner. The introduction starts with a qubit attached to a bath reaching thermalisation. Next, a system is attached to this qubit, and the author argues that the system should also thermalise. However, for this to happen, we need to be able to compute multi-time correlation functions for the qubit. This, the author claims, is not possible within the landscape of dynamical maps. A common workaround is to employ QRF, which the author shows to be fundamentally faulty (as is well known in the community, see [PhysRevA.90.022110,PhysRevA.106.022214]) and leads to inconsistent results.

The rest of the paper is dedicated to overcoming these deficiencies.

In Sec. 2, the author begins with the structure of multi-time correlations in quantum physics by introducing the notion of bitrajectories. The author is completely right about dynamical maps' inability to describe multi-time correlations; however, they can be generalised to deal with multi-time correlations using quantum combs or the process tensor, see PRXQuantum.2.030201. See arXiv:0712.0320, arXiv:1712.02589, arXiv:1802.03190,  and arXiv:2304.10258 on notions related to bitrajectories. Also, see a recent review on these methods at arXiv:2503.09693.

The ideas of Section 2 are not novel, they are also not trivial. The author does an excellent job of describing multi-time quantum correlations concisely. These ideas have been rediscovered a dozen times in the past 20 years, each for its own purpose, and the author adds to this growing list.

In Sec. 3, the author takes the ideas of Sec. 2 to a novel place. Multi-time correlations are messy, and it's difficult to categorise them. Usually, doing so requires operational motivations. Working off the problem defined in Sec. 1, the author motivates constructs such as correlation time and cumulants. In particular, the author argues that the cumulant accounts for all irreducible temporal correlations.

Section 4 goes back to the original problem of QRF and when it is valid in terms of the cumulants. This is already a very welcome contribution. The author goes on to further study the Born-Markov approximation using cumulants in Section 5. Finally, Section 6 goes beyond QRF. In Section 7, the author goes back to the problem of thermalisation to end the paper. These latter sections are rather technical, and I didn't check the calculations in detail.  However, this is the first rigorous treatment of these long-standing problems that I have seen.

The paper is clearly written, invents a new framework, and tackles old problems successfully. I strongly recommend this paper. I do think that the author should make links to the process tensor formalism, as this community is looking at multi-time quantum correlations from many different perspectives, and this will also give this paper the exposure that it deserves. Namely, the connections between the standard open dynamics approach and the multi-time correlation approach are still rather loose, and the author is in a strong position to have a serious impact.

Finally, here's another reference on quantum stochastic thermodynamics where many-time correlations arise naturally:
https://global.oup.com/academic/product/quantum-stochastic-thermodynamics-9780192895585.

Requested changes

My guess is that Eq. (3.5) cannot be exact due to the structure of quantum Markov order, see PhysRevLett.122.140401. However, an approximate versions should hold due to npj Quant. Info 7, 149 (2021).

Recommendation

Publish (surpasses expectations and criteria for this Journal; among top 10%)

  • validity: top
  • significance: top
  • originality: top
  • clarity: top
  • formatting: perfect
  • grammar: perfect

Author:  Piotr Szankowski  on 2025-08-04  [id 5700]

(in reply to Report 1 on 2025-05-28)

Warnings issued while processing user-supplied markup:

  • Inconsistency: Markdown and reStructuredText syntaxes are mixed. Markdown will be used.
    Add "#coerce:reST" or "#coerce:plain" as the first line of your text to force reStructuredText or no markup.
    You may also contact the helpdesk if the formatting is incorrect and you are unable to edit your text.

I would like to thank the Reviewer for their careful reading of the manuscript and for providing positive recommendations. Detailed responses to the Reviewers' questions and comments are given below.

Yours sincerely, P. Szańkowski

Comment 1

While the Introduction explains the problem with great clarity, the abstract does not. It says little about what is gained by computing multi-time correlations.

Answer: I have modified the abstract with a short list of topics where multi-time correlations play a crucial role.

Comment 2

It might help the reader to outline the structure of the paper at the end of the introduction.

Answer: I have added the outline at the end of the introduction.

Comment 3

In Sec. 2, the author begins with the structure of multi-time correlations in quantum physics by introducing the notion of bitrajectories. The author is completely right about dynamical maps' inability to describe multi-time correlations; however, they can be generalized to deal with multi-time correlations using quantum combs or the process tensor, see PRXQuantum.2.030201. See arXiv:0712.0320, arXiv:1712.02589, arXiv:1802.03190, and arXiv:2304.10258 on notions related to bitrajectories. Also, see a recent review on these methods at arXiv:2503.09693.

[...] I do think that the author should make links to the process tensor formalism, as this community is looking at multi-time quantum correlations from many different perspectives, and this will also give this paper the exposure that it deserves.

Answer: I thank the Reviewer for this comment. While developing the bi-trajectory language, we realized some time ago that there might be some links between the formalism we were working on and the process tensor approach. Soon after, we concluded that this link is real and it most likely runs much deeper than we initially thought. In Quantum 8, 1447 (2024) we wrote about our preliminary findings about these relations; it is clear to me that there is much more to this story. At the time of writing this paper it did not feel right to me to point out the fundamental connections between combs and bi-trajectories, without first researching the topic in more detail. Moreover, it seemed at this time that the comb connection might be the next full project we will be doing with my co-workers—it would be a waste to spoil our future findings with half-baked preliminary conclusions we had so far, I thought. (Plans got delayed, but we are still very much interested in pursuing this topic.) Of course, now I see that omitting quantum combs for such reasons was a silly mistake. To remedy this I have added a comment (and relevant references) in the opening paragraph of Sec. 2.2:

Multi-time correlation functions are ostensibly dynamical quantities. As such, the formal description of MTCs in terms of the standard state-focused formulation of quantum mechanics---which is tailored for describing `quantum statics' (i.e., single-time MTCs, or simply, expectation values of observables)---is a sub-optimal choice. Recently renewed interest in contexts extending beyond the standard quantum statics has stimulated development of powerful tools such as process tensors (or quantum combs) [Milz_Quantum2020, Milz_PRXQuantum2021, Taranto_arxiv2025], that can efficiently handle various multi-time objects encountered in quantum mechanics. Here, our method of choice is the so-called bi-trajectory formalism; it was developed previously in [Szankowski_SciRep2020, Szankowski_PRA2021, Szankowski_Quantum2024, Lonigro_Quantum2024, Szankowski_arXiv2024] and it is a natural fit for the specific context of open system MTCs. The bi-trajectory theory is a reformulation of quantum mechanics that shares some of its DNA with other trajectory-focused formalisms [Caves_PRD1986, Aharonov_PRA2009, Strasberg_PRX2024]. Moreover, a limited preliminary investigation [Lonigro_Quantum2024] suggests a deeply rooted connection between bi-trajectory formalism and process tensors, albeit the full understanding of this potentiality requires further study.

Comment 4

My guess is that Eq. (3.5) cannot be exact due to the structure of quantum Markov order, see PhysRevLett.122.140401. However, an approximate versions should hold due to npj Quant. Info 7, 149 (2021).

Answer: This comment seems to suggest it cannot be true that correlation functions/cumulants have finite range (and thus, finite correlation time) because typical quantum systems have infinite Markov order. If I am not misreading its meaning, the comment conflates the correlation time with the Markov order (understood, loosely, as the number of past conditions influencing the future). The implication is that short correlation time should correspond to a small number of past conditions, and long correlation time to a large number of conditions; in particular, Markovian processes (Markov order equal 1) would have to have shorter correlation times (zero?) than non-Markovian processes (Markov order greater than 1).

This, however, is incorrect: in reality, the range of correlations/cumulants is independent of the notion of Markovianity, Markov order, or other related concepts. It is a very common misconception that they are connected (or even the same), and, sadly, the literature is plagued by it. I think the unfortunate choice to dub Markovian processes as "memoryless" is largely to blame. Also, it does not help that the most common way to construct Markovian noise is to drive an otherwise deterministic system with a zero-correlation time white noise—this, I think, can inadvertently create a strong association in people’s minds that short correlation time leads to Markovianity, even though the two properties belong to two different processes (the driver is white but can be non-Markovian, cf. Phys. Rev. E 50, 2668 (1994) and the driven is Markovian but is non-white).

I find the following rule of thumb to be helpful in understanding the difference between Markovianity-related concepts and the correlation time: the "memorylessness" (i.e., Markovianity etc.) tells us how the process is "made", while correlations/cumulants tell us how the process "looks like from the outside". For example, in the context of open systems, the dynamics of the system–environment coupling operator is what directly influences the system. Therefore, the influence from the environment is encapsulated by correlation functions of the coupling. Since, in general, the same correlation functions can be produced by different underlying dynamical laws governing the environment, it is irrelevant whether the environment can or cannot be considered as Markovian.

Also notice that for Markovian processes (or a process with larger Markov order), the time-scales associated with the past conditions are completely irrelevant: the timings of the events constituting the process history do not matter, the only thing the Markovian process cares about is the latest event. Even if past events are clustered infinitesimally close to each other, or if the last event in the sequence happened in the distant past, Markovian process always depends only on the latest event. This focus on the relative chronology over the actual timing of past conditions only underlines the disconnect between Markovianity (Markov order in general) and time-scales associated with the process dynamics. Of course, in concrete calculations, the timings of the past conditions does affect the value of correlation functions and other derived quantities, regardless if the process is Markovian or not.

A classic example illustrating these points is a deterministic process describing a planet orbiting its sun: the process representing planet's position is Markovian (it is governed by time-local equations of motion), but the correlation time is infinite (or undefined, depending on one’s preference) because correlation functions of position are practically periodic.

It is not difficult to construct an analogous quantum model: Typical arrangement constitutes a system with discrete spectrum, $\hat H = \sum_a h(a)|a\rangle\langle a|$ and stationary initial state, $\hat\rho_0 = \sum_a p(a)|a\rangle\langle a|$. The system is Markovian by any metric: (i) its evolution $\hat U_{t,s} = {\exp}[-\mathrm i(t-s)\hat H]$ is CP divisible, $\hat U_{t,s}\hat U_{s,0} = \hat U_{t,0}$; (ii) quantum regression is exact, etc. Nevertheless, the correlation time is infinite (or undefined), as correlations of system’s observables are oscillatory, e.g.,

$$\operatorname{Re}\operatorname{tr}[\hat X(t)\hat X(t+\Delta t)\hat \rho_0] = \sum_{a,b}p(a)|X_{ab}|^2\cos[(h(a)-h(b))\Delta t]$$

However, when the system is complex, so that its spectrum becomes dense and degenerate, the revivals of the oscillations in correlations of some observables are getting more and more delayed. At some point, if this delay is long enough, one can treat correlation as a decaying function with finite correlation time (Phys. Rev. A 100, 052115 (2019))—this is the standard physical argument why environments like thermal baths should have finite-range correlation functions. Notice that the system remains Markovian: the Hamiltonian gets more complex, but it still generates divisible evolution.

In summary, Eq.(3.5) is inexact in the sense that it assumes that the recurrence time is so much longer than any other time-scale so that the system's correlations/cumulants can be treated as decaying rather than oscillatory functions. This assumption applies to sufficiently complex systems and its validity is unaffected by the Markov order characterizing the system.

---

## Round 2 · Referee Report · Anonymous (Referee 2) · 2025-7-22

Report

The manuscript concerns the analytical investigation of multi-time correlations in open quantum systems beyond the standard treatment, i.e., beyond the regime of validity of the quantum regression formula (QRF). Building on the bi-trajectory approach introduced by the author with coworkers in [19-23], here a perturbative approach is put forward essentially by rearranging and truncating a cumulant expansion in terms of the correlation strength among interventions at different times, as well as applying the standard Born-Markov approximation. Such an expansion reproduces the QRF at the lowest order, while the first order allows one to correctly predict the “propagation of thermalization”, i.e., that, given a system in contact with a thermal bath at temperature T, not only the system itself, but also a system in contact with it will thermalize at the same T.
The topic of the manuscript is timely and of interest within all those areas of research dealing with open quantum systems beyond the standard treatment (i.e., Lindblad master equation, quantum regression formula and so on), especially in view of the very few results in the literature that can be applied in a model-independent way to evaluate multi-time correlation functions in general regimes. The approach put forward here is novel and promising -- as shown by the application to the conceptually relevant problem of the propagation of thermalization – and it keeps quite a clear physical meaning. In particular, the bi-trajectory approach, along with the exploited cumulant expansion, allow the author to give an explicit and quantitative representation to the common intuition that a finite correlation time within the environment will imply a reduced number of interventions influencing each other.
Hence, I do think that the manuscript deserves publication in SciPost Physics, after the author will have taken into account the following remarks.
- The manuscript is essentially missing an actual introduction about multi-time correlation functions in open quantum systems, the quantum regression formula, and also the attempts to go beyond it – as said, there are not many, but indeed there are some: as an incomplete list of examples, the author can consider the following ones:
D. Alonso and I. de Vega, Multiple-time correlation functions for non-Markovian interaction: Beyond the quantum regression theorem, Phys. Rev. Lett. 94, 200403 (2005);
H.-S. Goan, P.-W. Chen, and C.-C. Jian, Non-Markovian finite-temperature two-time correlation functions of system operators: Beyond the quantum regression theorem, J. Chem. Phys. 134, 124112 (2011);
A. Ivanov and H.-P. Breuer, Extension of the Nakajima-Zwanzig approach to multitime correlation functions of open systems, Phys. Rev. A 92, 032113 (2015);
D. P. S. McCutcheon, Optical signatures of non-Markovian behavior in open quantum systems, Phys. Rev. A 93, 022119 (2016);
M. Ban, S. Kitajima, and F. Shibata, Two-time correlation function of an open quantum system in contact with a Gaussian reservoir, Phys. Rev. A 97, 052101 (2018).

I thus recommend to include an actual introduction with the content specified above and some references (those above, and/or others that can be useful) and only after that to start with the argument about the propagation of thermalization, clarifying that it is a – interesting and physically relevant – case study to clarify the limits of applicability of the QRF; this part might even be written in a separated section.
- At page 7, after equation 2.13, the term “discrete-time combs” appears, but without clarifying what the name “combs” is referring to. In view of the widespread use of the term “comb” within the community of open quantum systems – see e.g. Phys. Rev. Lett. 101, 060401 (2008) or the recent review arXiv:2503.09693, it should be clarified whether the name “comb” is referring to the same objects and the differences/peculiarities should be specified.
- Does the notation in the second equation in (5.12) mean that Lambda_{t’,t’} is the identity map? Maybe a more explicit notation or at least a clarification of this would help the reader.
- According to the general criteria of SciPost Physics, the conclusions of the manuscript are missing the “perspectives for future work”.
- A full stop is missing in equation (A3).

Recommendation

Ask for minor revision

  • validity: -
  • significance: -
  • originality: -
  • clarity: -
  • formatting: -
  • grammar: -

Author:  Piotr Szankowski  on 2025-08-04  [id 5701]

(in reply to Report 2 on 2025-07-22)

Warnings issued while processing user-supplied markup:

  • Inconsistency: Markdown and reStructuredText syntaxes are mixed. Markdown will be used.
    Add "#coerce:reST" or "#coerce:plain" as the first line of your text to force reStructuredText or no markup.
    You may also contact the helpdesk if the formatting is incorrect and you are unable to edit your text.

I would like to thank the Reviewers for their careful reading of the manuscript and for providing positive recommendations. Detailed responses to the Reviewers' questions and comments are given below.

Yours sincerely, P. Szańkowski

Comment 1

The manuscript is essentially missing an actual introduction about multi-time correlation functions in open quantum systems, the quantum regression formula, and also the attempts to go beyond it — as said, there are not many, but indeed there are some [...].

I thus recommend to include an actual introduction with the content specified above and some references (those above, and/or others that can be useful) and only after that to start with the argument about the propagation of thermalization, clarifying that it is a — interesting and physically relevant — case study to clarify the limits of applicability of the QRF; this part might even be written in a separated section.

Answer: I sympathize with the sentiment expressed in this comment; the structure of the paper does indeed deviate from the standard pattern. In fact, the initial drafts of this paper were written along such, more traditional, lines: The Introduction had a form suggested in the comment, and in content it more-or-less mirrored the current abstract. Then, I followed with sections containing technical definitions of MTC, first in closed, and then in open systems. Actually, back then, the problem of thermalization propagation—its introduction and solution via just defined perturbation theory—was crammed into a single section at the end of the paper.

However, it became quite apparent to me—and my feelings were validated by the feedback I got—that the traditional pattern does not work for this paper in particular, and maybe even the topic in general. The issue is that the target audience of this kind of work (i.e., MTCs plus open systems) is composed of two, non-overlapping, camps. On the one hand, the language of MTCs is natural for people with background in field theories (e.g., condnsed matter physics, ultra-cold gases, quantum optics, etc.), for many of whom the open systems framework is rather foreign. On the other hand, we have the portion of the audience "lured" in by the open systems aspect, who typically come from quantum information-adjacent backgrounds, where it is the concept of MTCs that is rather unnatural. (It is quite remarkable how often I had to explain to my colleagues that MTCs and correlations between sequential measurement results are not the same thing.) Therefore, to construct a proper introduction, which would not alienate either part of the target audience, one cannot rely on the reader's familiarity with MTCs nor the concepts of the standard open system theory—in other words, not only you have to define all the elements (MTCs, dynamical maps, GKLS theorem etc.) but you also have to show examples how those things are useful. Integrating this kind of introduction with the narrative about thermalization and its propagation—which is a topic universally familiar to physicists of any background—is an efficient way to bestow the otherwise technical introduction with a fairly easy to follow structure that showcases the use case for every concept involved. Hopefully, I was able to make my case why I prefer to keep the structure of paper as it is now.

In regards to the references pointed out by the Reviewer: upon reflection, I agree it was a mistake on my part that I neglected to cite some of the previous works concerning extensions to quantum regression theorem, which is, of course, strongly linked with open system MTCs. I was familiar with a couple of those works (although, not all of them, thank you for bringing them to my attention), but I decided that the approach adopted by them diverges too much from what I was doing in my paper. However, now I see that this difference in approach is a good point in and of itself—I should have reevaluated my assessment. In response, I have added the following paragraphs to the conclusions:

Previous works—e.g., [deVega_PRL2005,deVega_PRA2006,deVega_PRA2007,McCutcheon_PRA2016]—have proposed approximation schemes to improve on the QRF method. The approach adopted in these works is, in a sense, complementary to ours. The standard strategy is to address the problem of MTCs by extending the so-called quantum regression theorem (QRT) beyond the zero-correlation-time regime. Essentially, the QRT states that when $\tau^e \to 0$, the single- and two-time correlations of system observables satisfy the same dynamical equation [McCutcheon_PRA2016]; hence, QRT implies the exactness of QRF for zero-range cumulants, which is, of course, consistent with our theory. "Extending QRT" then amounts to writing down the derivative of a two-time MTC with respect to the second time argument and proposing an approximation that turns the expression into a legitimate dynamical equation—that is, closing the equation by reassembling the MTC on the right-hand side. Such an approximation is necessary because, as per QRT, the equation does not close on its own when $\tau^e > 0$.

It follows that theories utilizing this dynamical-equation approach result in MTC approximations analogous in form to the QRF method, where there are no explicit connections reaching across interventions. Instead, the deployed approximations lead to effective propagators between interventions that attempt to capture the effects that would result from cross-intervention connections, were they actually included. In contrast, our approach facilitates such cross-intervention connections, thereby allowing for the independent treatment of the in-between propagators, as demonstrated in Sec.5. Seemingly, there is no decisive argument favoring our approach over the traditional equation-of-motion method. When faced with a concrete problem, one must carefully consider the pros and cons of each method, and which approach is preferable will most likely depend on the fine details. However, there is something to be said for how intuitive it is to build up corrections to the MTC by progressively including more complex temporal correlations connecting the propagators across interventions—no such clear picture can be used to illustrate the physics of perturbation theory within the equation-of-motion framework.

Comment 2

At page 7, after equation 2.13, the term "discrete-time combs" appears, but without clarifying what the name "combs" is referring to. In view of the widespread use of the term "comb" within the community of open quantum systems — see e.g. Phys. Rev. Lett. 101, 060401 (2008) or the recent review arXiv:2503.09693, it should be clarified whether the name "comb" is referring to the same objects and the differences/peculiarities should be specified.

Answer: I have changed the name to "discrete-time filter function" to avoid confusion with quantum combs. Originally, I had "Dirac comb" in mind when I was naming the function $\Delta$; to be frank, I completely missed the conflict with process tensors/quantum combs. Thank you for pointing this out to me.

Comment 3

Does the notation in the second equation in (5.12) mean that $\Lambda_{t',t'}$ is the identity map? Maybe a more explicit notation or at least a clarification of this would help the reader.

Answer: Indeed, the standalone "bullet" $\bullet$ correspond to super-operator identity. I have added the line

where the super-operator identity is expressed using the "bullet" notation, $\bullet\hat A = \hat A$.

below Eq. (5.12) to clarify this notation.

Comment 4

According to the general criteria of SciPost Physics, the conclusions of the manuscript are missing the "perspectives for future work".

Answer: To address this issue I have added the following paragraph at the end of Sec. 8 (Conclusions):

Finally, we note that the hierarchy of correlation strength (3.6), which constitutes the basis for our theory, can be analyzed in terms of graphs. In this reading, one interpretation is to represent the moments of dynamical variables as the graph's vertices, and the edges connecting vertices as cumulants. Therefore, the introduced perturbation theory should naturally lend itself to some form of diagrammatic treatment. This opens a promising avenue for further development of the theory, with the prospect of a high payoff that is characteristic of diagrammatic methods known from other perturbation theories compatible with such an approach.

---

## Round 3 · Author Response

Aside the changes to the manuscript asked for by the Reviewers, I have corrected certain issues with Sec.1 (Introduction). First, some formulas in the original version were given in a simplified form that assumed in advance quantum regression formula; I have reverted them to their general, non-simplified form, see Eqs. (1.6), (1.7), (1.8), (1.10), (1.11), (1.12), and (1.16).
Second, I have made a mistake where I wrote that the Davies map, Eq.(1.6), will thermalize *any* initial state of the qubit $q$; this is untrue. In reality, $\hat\sigma_x$ is an eigenvector of the generator $\mathcal L^q$ corresponding to zero eigenvalue. Therefore, the dynamical map $\exp(t\mathcal L^q)$ will thermalize the initial state provided that $\operatorname{tr}(\hat\sigma_x\hat\rho_0^q) = 0$. In response, I have elected to set the initial condition in the example to a specific pure state $\hat\rho_0^q = |{1_z}\rangle\langle{1_z}|$. This mistake–and the subsequent fix–did not affected in any meaningful way the validity of the example.

---

## Round 3 · List of Changes

- Added a sentence to the abstract:
A complete characterization of the dynamics requires access to multi-time correlation functions as well: phenomena like detailed balance, linear and non-linear response, non-equilibrium transport in general, or even sequential measurements of system observables are all described in terms of multi-time correlations.
-
In Sec.1, several equations have been modified, see Eqs. (1.6), (1.7), (1.8), (1.10), (1.11), (1.12), and (1.16). For explanation, see the Author resubmission comment.
-
Added the paper outline at the end of Sec.1.
-
The notation for multi-time correlations has been slightly modified: $\langle A^+(t_2)B^-(t_1)\rangle_\rho \to \langle A(t_2^+)B(t_1^-)\rangle$.
-
The first paragraph of Sec.2.2: added a passage referring to process tensor formalism.
-
Changed the name of function $\Delta$: discrete-time comb -> discrete-time filter function.
-
Sec.3, Eq.(3.5): added a footnote explaining the difference between Markovianity and finite range of cumulants.
-
Sec.5, Eq.(5.12): added the clarification of the used notation for super-operator identity.
-
Sec.6, Eq.(6.4): fixed a slight error.
-
Sec.8: added a three paragraphs at the end of the section.

---

## Editorial Decision

published